# Quasi-solid-state Zn-air batteries with an atomically dispersed cobalt electrocatalyst and organohydrogel electrolyte

Qichen Wang[1,2,7], Qingguo Feng[3,7], Yongpeng Lei [1✉], Shuaihao Tang[4], Liang Xu[4], Yu Xiong [2], Guozhao Fang [5], Yuchao Wang[1], Peiyao Yang[1], Jingjing Liu[2], Wei Liu[6] & Xiang Xiong[1]

Quasi-solid-state Zn-air batteries are usually limited to relatively low-rate ability (<10 mA cm$^{-2}$), which is caused in part by sluggish oxygen electrocatalysis and unstable electrochemical interfaces. Here we present a high-rate and robust quasi-solid-state Zn-air battery enabled by atomically dispersed cobalt sites anchored on wrinkled nitrogen doped graphene as the air cathode and a polyacrylamide organohydrogel electrolyte with its hydrogen-bond network modified by the addition of dimethyl sulfoxide. This design enables a cycling current density of 100 mA cm$^{-2}$ over 50 h at 25 °C. A low-temperature cycling stability of over 300 h (at 0.5 mA cm$^{-2}$) with over 90% capacity retention at −60 °C and a broad temperature adaptability (−60 to 60 °C) are also demonstrated.

[1] State Key Laboratory of Powder Metallurgy, Central South University, Changsha, PR China. [2] Hunan Provincial Key Laboratory of Chemical Power Sources, College of Chemistry and Chemical Engineering, Central South University, Changsha, PR China. [3] Key Laboratory of Advanced Technologies of Materials, Ministry of Education, Southwest Jiaotong University, Chengdu, Sichuan, PR China. [4] Energy Materials Computing Center, Jiangxi University of Science and Technology, Nanchang, PR China. [5] School of Materials Science and Engineering, Central South University, Changsha, PR China. [6] State Key Laboratory of Fine Chemicals, Department of Chemistry, School of Chemical Engineering, Dalian University of Technology, Dalian, PR China. [7] These authors contributed equally: Qichen Wang, Qingguo Feng. ✉email: lypkd@163.com

Zn-air batteries (ZABs), with the merits of high theoretical energy density (1086 Wh kg$^{-1}$), environmental friendliness and abundance of the zinc anode, have attracted tremendous recent attention[1–3]. Nevertheless, the critical issues for high-rate (≥50 mA cm$^{-2}$) and robust ZABs are as follows: (i) finding suitable bifunctional O$_2$ electrocatalysts for the air cathode and (ii) ensuring sufficient stability of the electrochemical interfaces. The discharging/charging process of the air cathode is highly dependent on the O$_2$ reduction/evolution reaction (ORR/OER), respectively[4,5]. Unfortunately, the kinetics of the O$_2$ redox reactions are sluggish, causing substantial overpotentials. Although expensive and scarce precious metals (Pt for ORR; Ru/Ir for OER) have been used to accelerate the O=O bond cleavage and formation processes, durable and highly efficient electrocatalysts are still urgently needed[6,7].

Single-atom catalysts (SACs) supported on nitrogen-doped carbon possess high electrical conductivity and a well-defined atomic structure with a tunable coordination environment, exhibiting a near 100% utilization of the atomic metal sites[8–11]. Among transitional metals (e.g., Fe, Co, Ni, and Mn), electrocatalysts with atomically dispersed Co sites are promising alternatives for improving reversible O$_2$ redox kinetics. High accessibility of the atomic metal active sites and fast charge transfer are necessary for high-rate ZABs, which may be realized by designing an electrocatalyst with a hierarchical porous nanoarchitecture. The local coordination environment around the central metal atoms significantly affects their electronic and geometric structures, determining the adsorption strength of oxygenated intermediates to the atomic metal sites[12–17]. Compared to a flat geometry, an electrocatalyst with a curved nanostructure may lead to a larger electric field enhancement and thus accelerate the reaction kinetics[18,19].

Instability of the electrochemical interfaces, which can lead to Zn dendrites, side reactions, corrosion, etc., greatly affects the cycling lifespan of high-rate quasi-solid-state ZABs and deserves further research attention[20,21]. Currently, the conventional alkaline hydrogel electrolyte causes severe passivation of the Zn anode, leading to irreversibility of Zn plating/stripping and limited cycling capability. In response, adjusting the Zn$^{2+}$ solvation sheath structure and suppressing the water activity may improve the reversibility of Zn plating/stripping. Furthermore, in extremely low temperatures (≤−40 °C), the charge transfer rate sharply decreases, leading to severe polarization and even battery failure[22–24]. An anti-freezing and compatible hydrogel electrolyte enables adaptability under extreme conditions[25,26]. However, limited efforts have been spent to exploit hydrogel electrolytes for both high-rate and temperature-adaptive ZABs. As an organic solvent with strong polarity, dimethyl sulfoxide (DMSO) can be used as an additive to modulate the H-bond network of aqueous electrolytes. An organohydrogel electrolyte synthesized the DMSO/H$_2$O binary solvent may provide an opportunity to improve the performance of quasi-solid-state ZABs.

Here we demonstrate a quasi-solid-state ZAB that exhibits a cycling current density of 100 mA cm$^{-2}$, a robust cycling stability (50 h at 100 mA cm$^{-2}$), and a wide operating temperature ranging from −60 °C to 60 °C. The highly wrinkled graphene creates a large charge gradient around the Co-N$_4$ sites, which helps to strengthen the adsorption of oxygenated intermediates. Moreover, by tailoring the H-bond network of the organohydrogel electrolyte with DMSO, we improve the interfacial stability and low-temperaturetolerance of the ZABs. Our work simultaneously achieves a high-rate cycling ability at 25 °C and a broad temperature adaptability (−60 to 60 °C), which may provide future directions for further quasi-solid-state ZAB development.

## Results

### Synthesis and characterization of Co SA-NDGs.

The synthetic procedure of atomically dispersed Co sites supported on nitrogen-doped graphenes (Co SA-NDGs) consists of electrostatic adsorption, hydrothermal treatment, subsequent pyrolysis and acid leaching; full details are given in the Methods section and the synthesis is depicted graphically in Supplementary Fig. 1. Scanning electron microscopy (SEM, Supplementary Fig. 2) shows that Co SA-NDGs possess a 3D interconnected porous architecture with a highly wrinkled surface, delivering a Brunauer-Emmett-Teller (BET) surface area of 396.2 m$^2$ g$^{-1}$. The small mesopore centered at about 2–4 nm is conducive to balance the electrolyte permeation and the ion diffusion[27]. The X-ray diffraction (XRD) pattern and Raman spectra suggest a lower graphitization degree of Co SA-NDGs (Supplementary Figs. 3 and 4)[9,16]. Meanwhile, a strong electron paramagnetic resonance (EPR) signal affirms the presence of plenty of defects (e.g., dangling bonds and vacant atom sites) (Supplementary Fig. 5). Transmission electron microscopy (TEM) images (Fig. 1a and Supplementary Fig. 6a) show that the graphene nanosheets are highly curved. The high-curvature carbon lattices (highlighted by yellow dots) and the wrinkled nanosheets are clearly observed for Co SA-NDGs in high-resolution TEM (HRTEM) images (Fig. 1b, Supplementary Fig. 6b, c). The ring-like selected area electron diffraction (SAED) pattern signifies the poor crystallinity. The nanosheet structure is further confirmed by atomic force microscopy (AFM) analysis with an average thickness of 3.24 nm (Supplementary Fig. 7). Imaging with aberration-corrected high angle annular dark-field scanning transmission electron microscopy (AC HAADF-STEM) distinctly detects the monodispersed bright dots (marked by yellow circles) at an atomic level (Fig. 1c), corresponding to the atomically dispersed Co atoms[9,18]. The elemental mapping images of Co SA-NDGs (Supplementary Fig. 6d) verify the uniform distribution of Co, N, C, and O species.

The electronic structure and local atomic configuration of Co SA-NDGs were studied by X-ray photoelectron spectroscopy (XPS) as shown in Supplementary Fig. 8. The N 1s spectrum (Fig. 1d) is divided into pyridinic-N (398.2 eV), Co-N$_x$ (399.1 eV), pyrrolic-N (399.9 eV), graphitic-N (401.1 eV) and oxidized-N (402.1 eV), respectively. The abundant defects and nanopores increase the density of N atoms at edge sites, resulting in the formation of a high percentage of pyridinic-N[28]. Compared with the N-doped graphenes (NDGs), the positive shift (~0.2 eV) of the C–N bond in the C 1s spectrum (Fig. 1e) for Co SA-NDGs is associated with a strong charge transfer effect between single Co atoms and curved NDGs supports. The decreased electron density of C atoms and the atomic Co-N$_x$-C interface effectively facilitate the adsorption of intermediates and tailor the reaction energy barrier[29]. The three main peaks are assigned to C=O, C–O–C and C–OH in the fitted O 1s spectrum. These hydrophilic oxygen-containing-groups enhance the three-phase contact (Supplementary Fig. 8c). The loading content of Co is determined as 0.14 wt% by inductively coupled plasma optical emission spectrometry (ICP-OES). Furthermore, the Co K-edge of Co SA-NDGs in X-ray absorption near-edge structure (XANES) exhibits an increased white-line intensity compared to that of Co foil and CoO (Supplementary Fig. 9a), suggesting the valence state of Co atoms is positive. The average oxidation state of Co species is about +2.19 (Supplementary Fig. 9b). The extended X-ray absorption fine structure (EXAFS) spectra display two peaks located at ~1.4 Å and 1.9 Å (Fig. 1f and Supplementary Fig. 9c), which can be assigned to Co–N and Co–C scattering paths, respectively[30,31]. The coordination number of Co–N is ~3.3 (Supplementary Table 1), suggesting that the dominant Co–N coordination in Co SA-NDGs is likely to be Co-N$_4$ coordination.

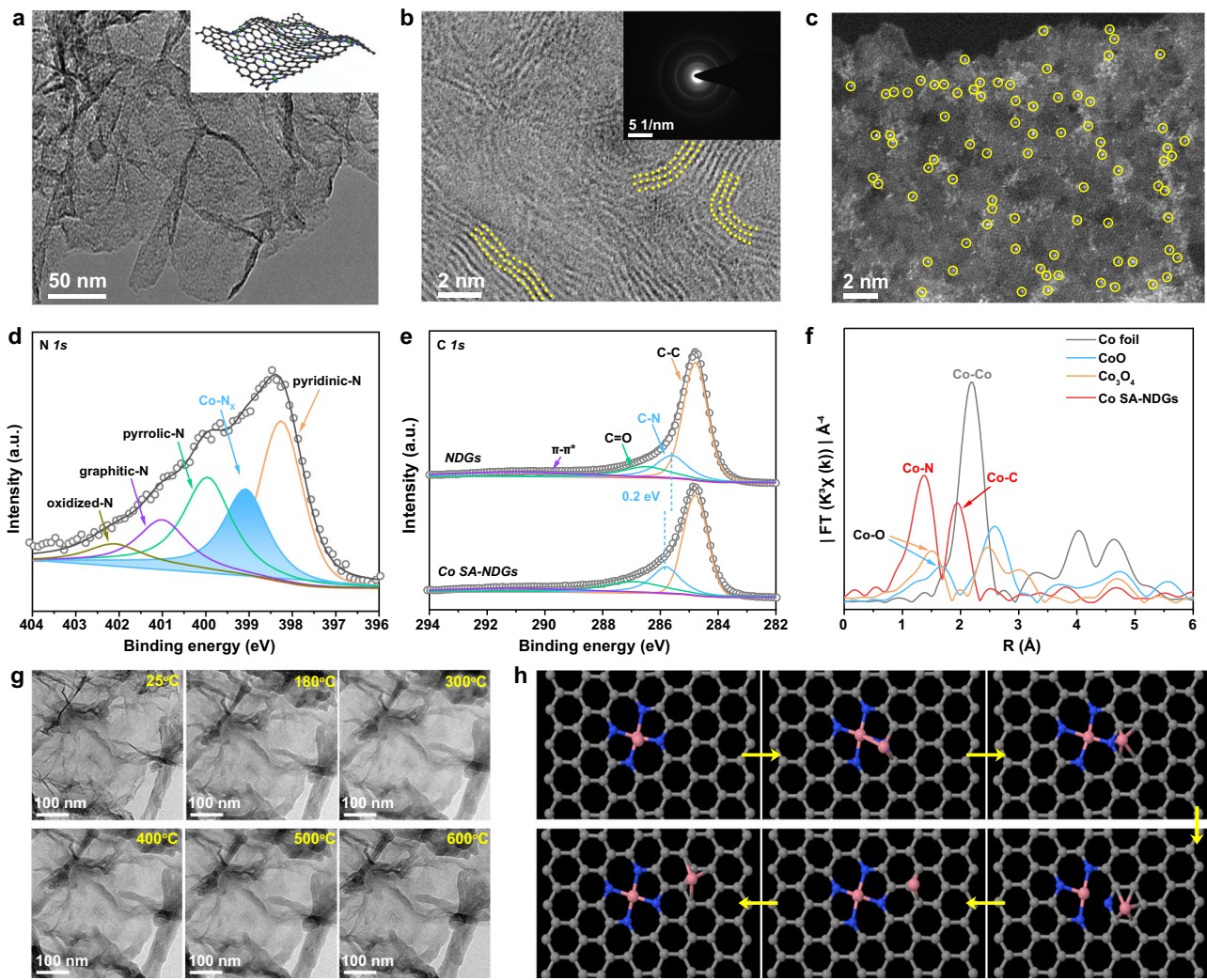

**Fig. 1 Structural characterizations and theoretical simulations of Co SA-NDGs. a** TEM image of Co SA-NDGs. Inset shows the corresponding curved model. **b** HRTEM image of Co SA-NDGs. Inset shows the corresponding SAED pattern. **c** AC HAADF-STEM image of Co SA-NDGs. **d** N *1s* spectrum of Co SA-NDGs. **e** C *1s* spectra of Co SA-NDGs and NDGs. **f** Fourier transform of EXAFS spectra of Co SA-NDGs, Co foil, CoO, and $Co_3O_4$ at the Co K-edge. **g** In situ temperature-dependent TEM images recorded at varying temperature. **h** Molecular dynamic simulations to reveal the dynamic evolution of Co dimer. The pink, blue, and gray balls refer to Co, N, and C atoms, respectively.

Besides, no peak of the metallic Co–Co bond or other high-shell peaks for Co SA-NDGs is detected compared with Co foil, confirming that the Co atoms are atomically dispersed.

To understand the evolution process of atomically dispersed Co atoms during pyrolysis, in situ temperature-dependent TEM investigations and molecular dynamics (MD) simulations were performed. The structural evolution was monitored in time with TEM in the temperature window ranging from 25 to 600 °C (Fig. 1g). In our case, the reduced graphene oxide (rGO) becomes less crystalline as the temperature rises to 600 °C, which may be ascribed to the heteroatom doping effect[32,33]. More interestingly, the related Co-based nanoparticles or clusters do not appear. Furthermore, based on our previous model[34], we placed a Co cluster consisting of two Co atoms on a tetrapyridine N-doped carbon site to study the possible changes at 800 °C. The results show that two Co single atoms do not form a stable Co cluster (Fig. 1h). The Co-Co bond of the cluster gradually becomes longer until it is broken. The Co connected to the tetrapyridine N automatically forms a typical planar $Co-N_4$ configuration, while the other Co atom will move further and further away from the $Co-N_4$ coordination. This result fully proves that stable Co dimers are not easy to form, but due to the synergistic effect of charge

transfer and vacancy defects, the typical planar four-coordinate configuration of $Co-N_4$ is formed. Simultaneously, the escaped Co atom may be captured by the doped N species or graphene defects until it is stabilized. The entire evolution process with more sets of images is supplemented (Supplementary Movie 1). These experimental observations and MD simulations provide solid evidences of the evolution process for the atomically dispersed $Co-N_4$ coordination during synthesis.

**Electrocatalytic $O_2$ performance and DFT calculations.** The $O_2$ electrocatalytic activities of Co SA-NDGs, NDGs and commercial Pt/C were comprehensively evaluated in 0.1 M KOH solution. Linear sweep voltammetry (LSV) curves (Fig. 2a) exhibit that Co SA-NDGs have an excellent ORR activity with an onset potential ($E_{onset}$) of 1.02 V vs. the reversible hydrogen electrode (RHE), superior to that of Pt/C (0.96 V). At the potential of 0.85 V, the Co SA-NDGs reach a kinetic current density ($j_k$) of 11.7 mA cm$^{-2}$ (Fig. 2b), ~2.18 times higher than that of Pt/C (5.36 mA cm$^{-2}$). The rapid ORR kinetic of Co SA-NDGs is supported by a small Tafel slope of 54 mV dec$^{-1}$ (Supplementary Fig. 10b). The fitted Koutecky-Levich (K-L) plots calculated from the LSV curves show

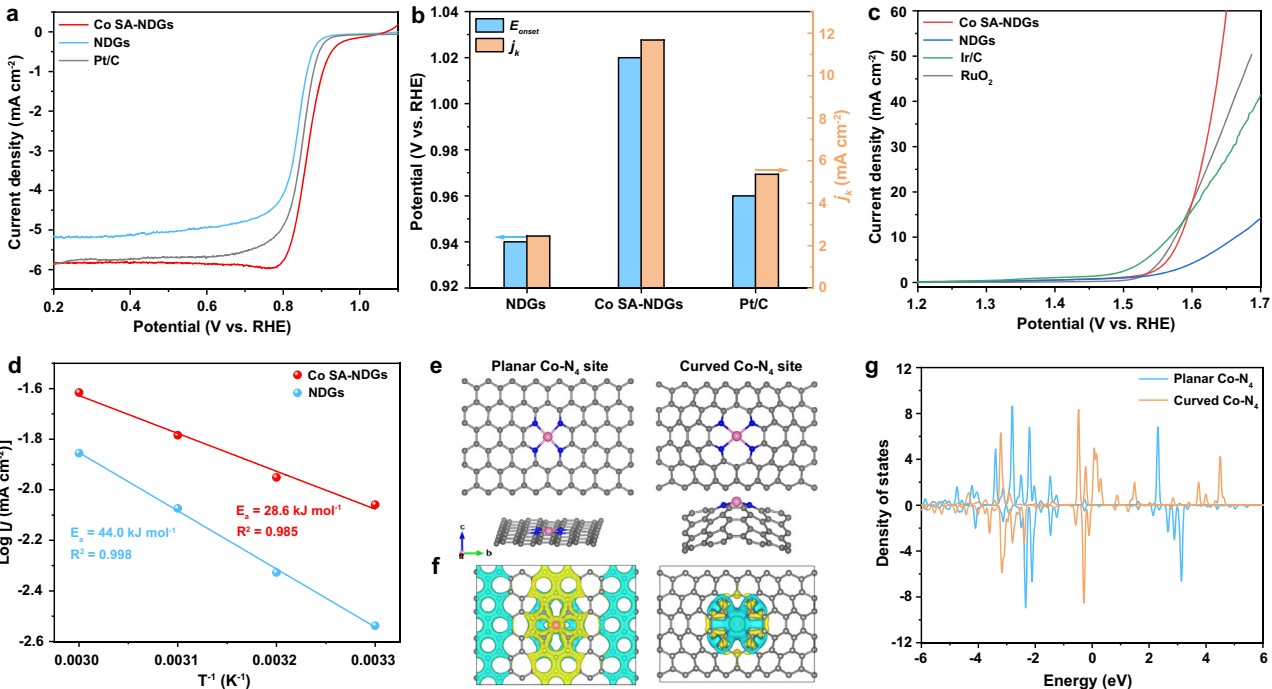

**Fig. 2 Electrocatalytic O2 performance and DFT calculations of Co SA-NDGs. a** LSV curves of Co SA-NDGs, NDGs and Pt/C catalysts for ORR.
**b** Comparison of $E_{onset}$ and $j_k$ at 0.85 V (V vs. RHE). **c** LSV curves of Co SA-NDGs, NDGs, Ir/C and RuO2 for OER. **d** Arrhenius plots of Co SA-NDGs and
NDGs at the overpotential of 350 mV. **e** Established models of planar Co-N4 site and curved Co-N4 site. **f** Calculated differential charge density. Yellow and
cyan areas represent charge density aggregation and depletion, respectively. **g** Density of states for planar Co-N4 site and curved Co-N4 site.

a good linearity (Supplementary Fig. 11), signifying that the Co SA-NDGs mainly follow the 4e− ORR pathway[35,36]. The accelerated degradation test of Co SA-NDGs was performed by cycling the catalyst between 0.6 and 1.0 V in over 5000 continuous cycles (Supplementary Fig. 12a), showing a small negative shift (11 mV) in the half-wave potential ($E_{1/2}$). After injecting methanol into the electrolyte, almost no disturbance in current response is found for Co SA-NDGs, whereas a sharp current decline is observed for Pt/C (Supplementary Fig. 12b). The Co species are still well distributed after the durability test, evidenced by TEM and mapping results (Supplementary Fig. 13). These results confirm good ORR stability and methanol tolerance of Co SA-NDGs.

Then, the electrocatalytic OER behavior of Co SA-NDGs was also assessed. An overpotential ($E_{10}$) of 350 mV for Co SA-NDGs is required to reach current density of 10 mA cm−2 (Fig. 2c), which is comparable to that of Ir/C (345 mV) and RuO2 (338 mV) but lower than that of NDGs (430 mV). Moreover, the rapid OER kinetics are elucidated by Tafel slope and electrochemical impedance spectroscopy (EIS) (Supplementary Fig. 14). Specifically, the activation energy ($E_a$), which is obtained from the slope of the plotted log ($j$) vs. 1/T, quantifies the energy barrier of a catalytic reaction[37]. The experimentally measured $E_a$ at an overpotential of 350 mV of Co SA-NDGs and NDGs are 28.6 and 44.0 kJ mol−1 (Fig. 2d and Supplementary Fig. 15), respectively, suggesting favorable OER kinetics of Co SA-NDGs. The electrochemical active surface area (ECSA) can be estimated from double layer capacitance ($C_{dl}$) because of a linear relationship[34,36]. The measured $C_{dl}$ values are 26.3 and 20.6 mF cm−2 for Co SA-NDGs and NDGs (Supplementary Fig. 16), corresponding to the ECSA of 657.5 and 515.0 cm2, respectively. The high ECSA of the Co SA-NDGs suggests a greater exposure of active sites compared to NDGs. Generally, the bifunctional activity is evaluated by ΔE (ΔE = $E_{OER:j=10}$−$E_{ORR:1/2}$). We represent the ORR $E_{1/2}$ and OER $E_{10}$ on the vertical and horizontal axes, respectively, for better visual comparison, where

the point located at the upper right demonstrates a better bifunctional capability[38]. The ΔE of Co SA-NDGs (0.71 V) is lower than selected non-precious metal SACs reported in recent literature (Supplementary Fig. 17 and Supplementary Table 2).

The crucial role of curvature on the atomic Co-N4-C system was studied using first-principles calculations based on DFT[39] (Fig. 2e). Figure 2f gives the calculated differential charge densities for the planar Co-N4 and curved Co-N4 model to show the charge transfer difference. The curved Co-N4 shows more localized charge densities and a larger charge gradient than the planar Co-N4, which is believed to facilitate the subsequent O2 activation[40]. Density of states (DOS) analysis (Fig. 2g) shows that the d-band center of curved Co-N4 is downshifted compared to that of planar Co-N4, indicating that a stronger adsorption of oxygenated intermediates will be achieved. We have performed a Bader charge analysis for our calculations[41]. It reveals that, when there are no molecules or radicals adsorbed, the Co atom will lose about 1.1 electrons for the curved structure and 1.08 electrons for the planar structure, while the neighboring N atom will gain about 2.676 and 2.575 electrons, respectively. Hence more electrons will be transferred from Co to N in the curved structure than that in the planar structure. When OH was adsorbed, Co atom will be more oxidized. The detailed data are supplied as Supplementary Table 3. Similar phenomena are observed when another radical is adsorbed on the active site. The free energy pathways of the planar and curved models are both downhill on the four-electron ORR processes at $U = 0$ V, revealing that each elementary step can be carried out spontaneously (Supplementary Fig. 18). At $U = 1.23$ V, the rate-determining step is the first protonation of the adsorbed OOH* species. The ORR overpotential of the curved Co-N4 site under alkaline condition is calculated to be 0.758 eV, which is lower than that of the planar Co-N4 site (0.90 eV), suggesting the beneficial effect of graphene curvature for optimizing the ORR activity. Meanwhile, the optimized adsorbed intermediates for the ORR pathway are

shown for both the planar and curved Co-N$_4$ geometries in Supplementary Fig. 19.

**Aqueous ZABs with Co SA-NDGs.** Then aqueous ZABs with Co SA-NDGs as the air cathode were assembled and compared to those assembled with Pt/C as the air cathode (Supplementary Fig. 20), in which the ambient air acted as the cathode active agent[42]. The open circuit voltage (OCV) and maximum power density of Co SA-NDGs-based aqueous ZABs are 1.53 V and 251.4 mW cm$^{-2}$ (Supplementary Fig. 21a, b), respectively, which are higher than these of benchmark Pt/C-based aqueous ZABs (1.48 V and 177.0 mW cm$^{-2}$). To confirm the crucial role of the hierarchically macro/mesoporous structure of Co SA-NDGs in facilitating rapid ion transfer, the control sample (denoted as control Co SA-NDGs) was prepared by the same procedure as that used for synthesis of Co SA-NDGs except without freeze-drying treatment. As shown in Supplementary Fig. 22, the control Co SA-NDGs shows a low BET surface area of 51.9 m$^2$ g$^{-1}$. The control Co SA-NDGs cathode displays a decreased discharge current density at high overpotential compared to the Co SA-NDGs cathode, indicating that the hierarchically macro/meso-porous structure can reduce concentration polarization. The Co SA-NDGs cathode delivers a high specific capacity of 757.4 mAh g$^{-1}$ and an energy density of 956 Wh kg$^{-1}$ (both calculated based on the mass of zinc) at 10 mA cm$^{-2}$ in Supplementary Fig. 21c, outperforming those of Pt/C-based counterpart (630.8 mAh g$^{-1}$, 788 Wh kg$^{-1}$). The rate capability is further assessed by comparing the discharge voltage profiles at a series of current densities (Supplementary Fig. 21d). The Co SA-NDGs cathode enables higher resultant discharge voltage plateaus compared to the Pt/C cathode especially at high discharge current densities over 50 mA cm$^{-2}$. The average discharge voltages are 1.28, 1.24, 0.99, 0.78, and 0.57 V for the Co SA-NDGs cathode at 20, 50, 100, 150, and 200 mA cm$^{-2}$, respectively, which are higher than these of the Pt/C cathode (1.25, 1.03, 0.57, 0.36, and 0.21 V). Notably, the ZABs with Co SA-NDGs could discharge more than 240 h at 100 mA cm$^{-2}$ (Supplementary Fig. 21e). The unstable interface induced by the generated Zn dendrites may account for the voltage loss at the initial stage (Supplementary Fig. 21f). Unlike the Pt/C + RuO$_2$-based ZABs with significant voltage decay, a robust charging/discharging stability (300 h, one charge/discharge cycle is 22 min) for Co SA-NDGs-based ZABs is achieved (Supplementary Fig. 23a). Even when the charging/discharging time of each cycle is extended to 1 h, the Co SA-NDGs-based aqueous ZABs still maintain a robust voltage profile without obvious polarization (Supplementary Fig. 23b). Conclusively, the performance (e.g., maximum power density, specific capacity and cycling stability) is comparable to other aqueous ZABs reported in the literature thus far (Supplementary Table 4), which mainly originates from atomically dispersed Co-N$_4$ sites with curved nanostructure and the hierarchically interconnected structure of the graphene architecture.

**Quasi-solid-state ZABs with Co SA-NDGs.** Here, we firstly synthesize and examine the physicochemical properties of the polyacrylamide (PAM) hydrogel (Supplementary Fig. 24). The quasi-solid-state ZABs with Co SA-NDGs using PAM hydrogel electrolyte show a maximum power density of 219.9 mW cm$^{-2}$ and a stable discharge voltage profile when varying discharge current densities (2, 5, and 10 mA cm$^{-2}$). The discharge voltage in the rate performance is higher than that in the cycling charging/discharging measurement at 5 mA cm$^{-2}$ (Supplementary Fig. 25b, c), which may be attributed to the interface difference[43,44]. However, the high-rate performance is still challenging because of the formation of dense Zn dendrites and the

destruction of Zn|hydrogel electrolyte interface (Supplementary Figs. 25 and 26). Therefore, modulating the intrinsic properties of the hydrogel electrolyte is particularly vital to alleviate these issues for achieving high-rate capability at ambient condition.

As known, DMSO is a favorable H-bond acceptor, forming a strong H-bond network with water molecules (Fig. 3a), reconstructing the solvation sheath structure of Zn$^{2+}$. On the one hand, the hydrogen evolution reaction activity is obviously suppressed by the addition of DMSO (Supplementary Fig. 27)[26,45–47]. On the other hand, in situ optical visualization observations reveal a smooth interface of Zn|electrolyte (Fig. 3b, c) when adding DMSO into electrolyte, whereas severe Zn dendrites appear in the DMSO-free electrolyte[48]. The PAM organohydrogel synthesized in DMSO/H$_2$O binary solvent systems retains a good stretchability (Supplementary Fig. 28). SEM reveals an interconnected porous structure (Fig. 3d), facilitating electrolyte trapping and fast Zn$^{2+}$ ion diffusion during electrochemical reactions. In Fig. 3e, the ionic conductivity of the PAM organohydrogel electrolyte shows a high conductivity at 20 °C (2.6 mS cm$^{-1}$) and even at −40, −60 and 60 °C (0.40, 0.087 and 475.6 mS cm$^{-1}$, respectively), illustrating an efficient broad temperature adaptability. In order to investigate the compatibility between the Zn anode and the organohydrogel electrolyte, Zn∥Zn symmetric cells were constructed. The Zn∥ Zn symmetric cell with the PAM organohydrogel electrolyte displays a lower polarization during plating/stripping cycles at 2 mA cm$^{-2}$ for 200 h in contrast to the PAM hydrogel electrolyte (Fig. 3f), suggesting a stable electrochemical interface. Meanwhile, ex situ XRD patterns display the relatively weak intensity of formed Zn dendrites (ZnO) when using the PAM organohydrogel electrolyte (Supplementary Fig. 29). These results indicate that the introduction of DMSO can effectively alleviate Zn dendrites and improve Zn|gel electrolyte interface stability via modulating the H-bond network of the organohydrogel electrolyte. Notably, the quasi-solid-state ZABs with Co SA-NDGs using the PAM organohydrogel electrolyte present a robust charging/discharging cycle at 50 and 100 mA cm$^{-2}$ (Fig. 3g). Moreover, the structural stability and integrity of the cathodes after 50 h of cycling at 100 mA cm$^{-2}$ are confirmed (Supplementary Figs. 30 and 31). A comparison to the performance of other quasi-solid-state ZABs reported in recent literature highlights this good result at high cycling rates, implying the promising application (Supplementary Fig. 32 and Table 5).

Furthermore, considering the practical demand of rechargeable batteries in cold regions, highland, etc., there is a need to develop low-temperature quasi-solid-state ZABs. As far as we know, ZABs operating below −40 °C is seldomly reported owing to the pronounced increase in interfacial and charge-transfer resistance when the operating temperature drops from 25 to −60 °C (Supplementary Fig. 33). The slow ion transport in low-temperature (≤ 0 °C) environments limits the depth of discharge and leads to low critical current densities[49]. Then, the temperature-tolerance abilities of the PAM organohydrogel electrolyte are further rationalized by differential scanning calorimetry (DSC) and dynamic mechanical analysis (DMA)[50,51]. As shown in Fig. 4a, the PAM organohydrogel electrolyte remains transparent at −40 °C and further transforms into an opaque slurry gel at −60 °C. The DSC curve shows that the freezing point of the PAM organohydrogel electrolyte is less than −70 °C without the emergence of an exothermic peak. Apart from the strong inter-molecular H-bonds between DMSO and H$_2$O, the binding energy (E$_b$) between water and the terminal group of PAM organohydrogel electrolytes also contributes to lower the solid–liquid transition point[21,52–54]. In Fig. 4b, although the E$_b$ (−0.198 eV@PAM-W) of the terminal acylamino group with neighboring water molecules via the dipole-dipole

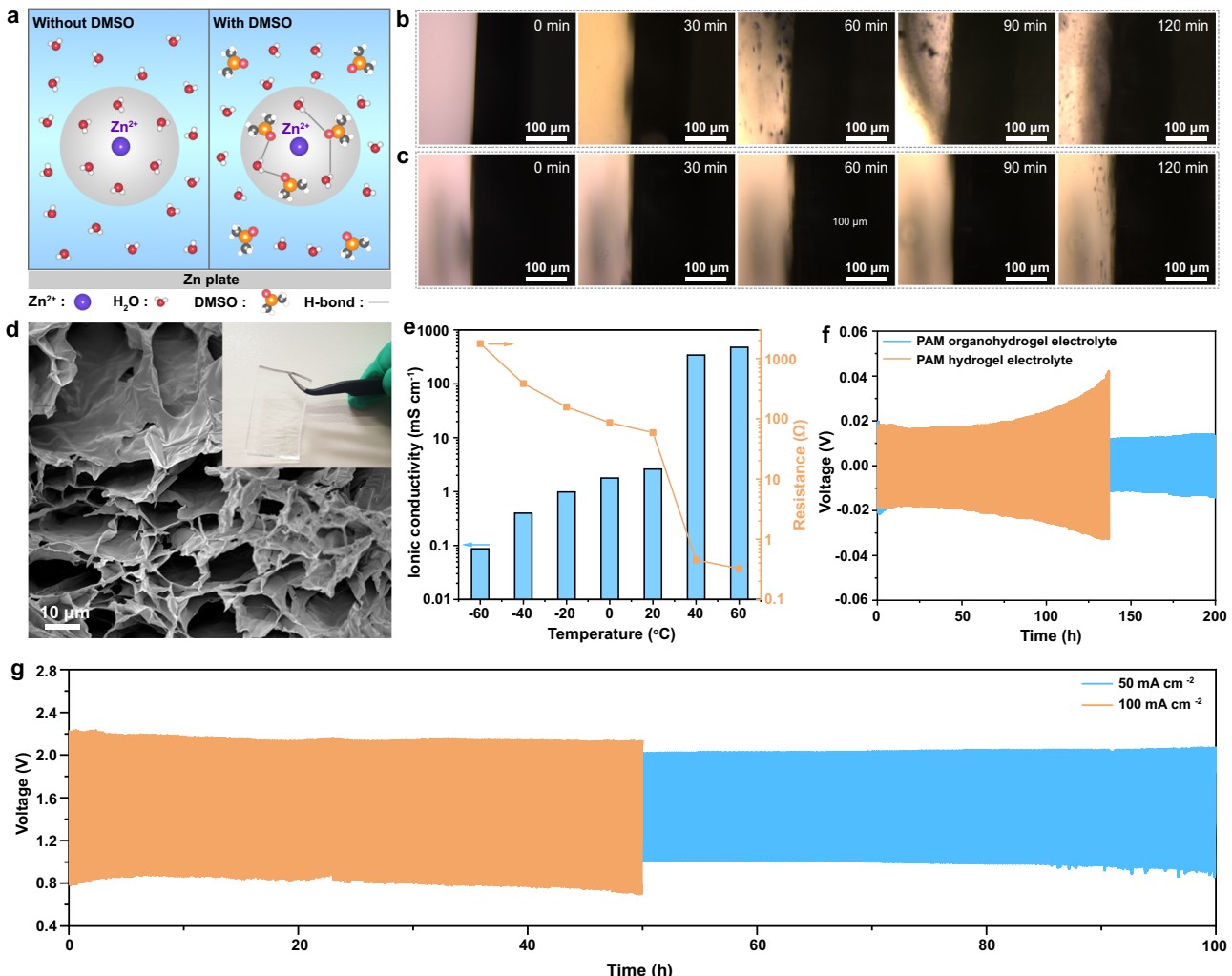

**Fig. 3 The role of DMSO in aqueous electrolytes, characterizations of the organohydrogel electrolyte and electrochemical performance of the Co SA-NDGs-based quasi-solid-state ZABs. a** $Zn^{2+}$ solvation structure and formed H-bond between DMSO and $H_2O$ molecules. The gray circles correspond to the solvation sheath structure of $Zn^{2+}$. In situ optical visualization observation of the Zn|electrolyte interface during different operating conditions (charging/discharging cycle time of 10 min): **b** charging/discharging at 5 mA cm$^{-2}$ in 6 M KOH + 0.2 M Zn(Ac)$_2$; **c** charging/discharging at 5 mA cm$^{-2}$ in 6 M KOH + 2 M DMSO + 0.2 M Zn(Ac)$_2$. **d** SEM image of the freeze-dried PAM organohydrogel electrolyte. Inset shows the corresponding optical photograph of PAM organohydrogel. **e** Ionic conductivity for PAM organohydrogel electrolyte at various operating temperatures. **f** Cycling performance of the symmetric Zn||Zn cells employing PAM hydrogel electrolyte and PAM organohydrogel electrolyte at 2 mA cm$^{-2}$. **g** Charging/discharging performance of quasi-solid-state ZABs with Co SA-NDGs at 50 and 100 mA cm$^{-2}$.

interaction is slightly stronger than that of two water molecules ($-0.171$ eV@W-W), the $E_b$ of the alkalified acylamino group with a water molecule (A-PAM-W) changes substantially to $-0.344$ eV. In principle, the stronger $E_b$ manifests as an efficient the anti-freezing properties of hydrogels electrolyte at low-temperature environments. The symmetric Zn||Zn cell with the PAM organohydrogel electrolyte exhibits a stable Zn plating/stripping process over 500 h at $-60\,°C$ (Supplementary Fig. 34). A rough surface and dendrite-free morphology of the cycled Zn plate (Fig. 4c) contribute to cycling stability[55]. Additionally, the strong H-bonds between DMSO and $H_2O$ result in a decrease of $H_2O$ molecule saturated vapor pressure, preventing $H_2O$ from evaporating at elevated temperatures ($\geq 40\,°C$). The DMA result shows that the glass transition temperature ($T_g$) of the PAM organohydrogel is $125\,°C$ (Supplementary Fig. 35). These results are indicative of anti-freezing and thermally stable properties for the as-synthesized PAM organohydrogel electrolyte.

At $-40\,°C$ (Supplementary Fig. 36), the critical current density of the assembled quasi-solid-state ZABs is 2 mA cm$^{-2}$ under the

steady-state discharge test. The specific capacity still reaches 778.4 mAh g$^{-1}$ at 2 mA cm$^{-2}$ and $-40\,°C$, corresponding to an energy density of 918.5 Wh kg$^{-1}$. A comparison between the energy density and operating temperature of the fabricated quasi-solid-state ZABs and other low-temperature batteries previously reported (Supplementary Table 6), suggests the intrinsic advantage of ZABs for low-temperature energy storage. When further decreasing the temperature to $-60\,°C$, the discharge voltages at 0.1, 0.5, and 1.0 mA cm$^{-2}$ are 1.302, 1.245, and 1.190 V (Fig. 4d), respectively. Fig. 4e exhibits the charging/discharging cycles at 0.5 and 1.0 mA cm$^{-2}$. The robust durability while retaining over 90% of its initial capacity is achieved at $-60\,°C$. To the best of our knowledge, this is the lowest operation temperature of ZABs reported so far. Moreover, this quasi-solid-state ZABs also operate well at temperatures ranging from 20 to 60 °C (Supplementary Fig. 37). At an elevated temperature of 60 °C, the maximum power density is 285.7 mW cm$^{-2}$ with the average discharge voltages of 1.26 V at 10 mA cm$^{-2}$ and 1.23 V at 20 mA cm$^{-2}$. The voltage losses are 0.046 and 0.214 V at 10 and

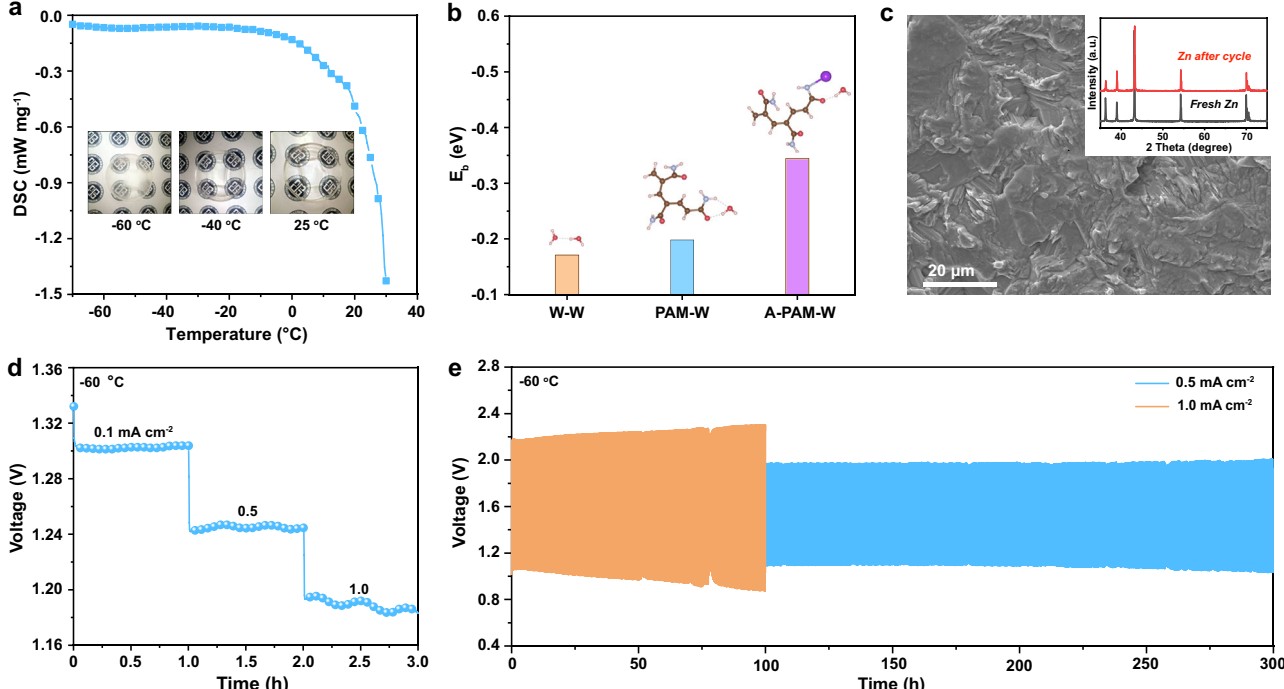

**Fig. 4 Characterizations of the organohydrogel electrolyte and electrochemical performance of the Co SA-NDGs-based quasi-sold-state ZABs at −60 °C. a** DSC curve of the PAM organohydrogel electrolyte. Inset shows the corresponding photographs recoded at 25, −40, and −60 °C. **b** The $E_b$ of W-W, PAM-W, and A-PAM-W. The purple, brown, lavender, red, and pink balls refer to K, C, N, O, and H atoms, respectively. **c** SEM image of Zn plate after 500 h cycling test of symmetric Zn||Zn cell at 0.5 mA cm$^{-2}$ and −60 °C. Inset shows the corresponding XRD patterns. **d** Rate performance of the Co SA-NDGs-based quasi-solid-state ZABs at −60 °C. **e** Charging/discharging cycling performance of quasi-solid-state ZABs with Co SA-NDGs at different current densities at −60 °C.

20 mA cm$^{-2}$, respectively, verifying its good cycling stability after 60 h. These results confirm the broad temperature adaptability of quasi-solid-state ZABs in this work.

## Discussion

In summary, high-rate quasi-solid-state ZABs were constructed. On the one hand, the curvature enhances the $O_2$ adsorption on Co-$N_4$ sites and strengthens the adsorption of oxygenated intermediates. Also, the highly interconnected graphene network with a hierarchical porous structure of Co SA-NDGs is conducive to fast charge transport and high-rate cycling. On the other hand, the PAM organohydrogel electrolyte mitigates Zn dendrite formation and suppresses side reactions, significantly increasing the electrochemical interface stability via modulating the cationic solvation structure. Consequently, at cycling rates as high as 100 mA cm$^{-2}$, the quasi-solid-state ZABs achieve a robust stability of 50 h. Benefiting from the depressed freezing point and increased thermal stability of the PAM organohydrogel electrolyte, the quasi-solid-state ZABs still show robust cycling stability with high-capacity retention at −60 and 60 °C.

## Methods

**Chemicals.** Co(NO$_3$)$_2$·6H$_2$O (AR), polytetrafluoroethylene (60 wt%), acetylene black (99%) and dimethyl sulfoxide (AR) were purchased from Macklin. Acrylamide (99%), N,N′-methylenebisacrylamide (99%), potassium persulfate (99.99%) were purchased from Aladdin. Methanol (AR), isopropanol (AR), potassium hydroxide (AR), hydrochloric acid (AR), and Zn(Ac)$_2$·2H$_2$O (AR) were purchased from Sinopharm Chemical Reagent Co., Ltd, China. Zn plate was purchased from Shanghai Maikelong Co., Ltd. Commercial Pt/C (20 wt%), Pt/C + RuO$_2$ (30 wt%) electrocatalyst and Nafion solution (5 wt%, D520) were purchased from Shanghai Hesen Electric Co., Ltd. All chemicals were from commercial sources and used without further purification. Water-proof breathable membrane and Ni foam were purchased from Changsha spring new energy technology Co., Ltd. Graphene oxide aqueous solution and g-C$_3$N$_4$ nanosheets are synthesized by the method we previously reported[34]. N$_2$ (99.99%) and O$_2$ (99.99%) were purchased from Changsha

Xinxiang Gas Chemical Co., Ltd. Milli-Q water (18.2 MΩ) was obtained from milli-Q system (UPT).

**Material synthesis.** For the synthesis of Co SA-NDGs, 20 mg of Co(NO$_3$)$_2$·6H$_2$O was added into 30 mL of graphene oxide aqueous solution (2 mg mL$^{-1}$) for ultrasonic dispersion of 1 h. Then, 300 mg of g-C$_3$N$_4$ nanosheets were added into the above solution with vigorous stirring at 1000 r min$^{-1}$ for 2 h at 25 °C. This precursor solution was transferred into an autoclave (50 mL) and then kept at 180 °C for 12 h. After cooling, a black hydrogel was freeze-dried for 72 h to remove excess H$_2$O while maintaining its pristine morphology. The dried cylindric aerogel was placed in a tube furnace and then heated to 800 °C (3 °C min$^{-1}$) for 3 h under flowing N$_2$ protection, followed by cooling to 25 °C. The as-prepared sample was etched in 1 M HCl solution at 80 °C for 48 h. The sample was subsequently washed with H$_2$O for five times and finally vacuum-dried at 60 °C for 24 h. NDGs was prepared with the same synthesis procedure of Co SA-NDGs except that without Co(NO$_3$)$_2$·6H$_2$O introduction.

**Synthesis of the PAM hydrogel electrolyte.** The stretchable gel electrolyte was fabricated by a free radical polymerization approach. The function of the initiator is to promote the formation of free radicals and initiate polymerization. The PAM hydrogel electrolyte was prepared as follows: 4 g of acrylamide (AM), 4 mg of N,N′-methylenebisacrylamide (MBAA, Crosslinking agent), and 10 mg of potassium persulfate (K$_2$S$_2$O$_8$, Initiator) were dissolved in 10 mL of Milli-Q water under magnetic stirring at 500 r min$^{-1}$. The resultant solution was poured into a transparent box (Length, width and height are 8, 8, and 5 cm, respectively). Then the box was sealed with tape and placed in an oven at 60 °C for 12 h. The as-prepared PAM hydrogel (thickness of 0.3 cm) was immersed in the 6 M KOH + 0.2 M Zn(Ac)$_2$·6H$_2$O electrolyte for 72 h before use.

**Synthesis of the PAM organohydrogel electrolyte.** The PAM organohydrogel electrolyte was prepared as follows: 4 g of AM, 4 mg of MBAA, and 10 mg of K$_2$S$_2$O$_8$ were dissolved in 10 mL mixed solvent (5 mL of Milli-Q water and 5 mL of DMSO) under magnetic stirring at 500 r min$^{-1}$. The resultant solution was poured into a transparent box. Then the box was sealed with tape and placed in an oven at 60 °C for 12 h. The as-prepared PAM organohydrogel (thickness of 0.3 cm) was immersed in the 6 M KOH + 0.2 M Zn(Ac)$_2$·6H$_2$O electrolyte for 72 h before use.

**Material characterization**. Field-emission SEM (Hitachi S-480, Japan) was performed to investigate the morphology. TEM, high-resolution TEM (JEOL JEM-2100F, Japan) and elemental mappings were conducted to characterize microstructure. High-angle annular dark-field scanning transmission electron microscopy (HAADF-STEM) images were collected by using a JEOL 200 F transmission electron microscope operated at 200 keV, equipped with a probe spherical aberration corrector. Atomic force microscopy (AFM) was carried out on a Nanonavi E-Sweep N environment control scanning probe microscope. X-ray diffraction (XRD) patterns were recorded on a Bruker AXS D8 Advance device using Cu-Kα radiation (λ = 1.5418 Å). Raman spectra were collected on a Renishaw in Via Raman spectrometer with a laser wavelength of 532 nm. XPS experiments were conducted on a RBD upgraded PHI-5000C ESCA system (Perkin Elmer) with Mg Kα radiation (h = 1253.6 eV). The Brunauer–Emmett–Teller (BET) specific surface area was measured by nitrogen adsorption at 77 K on a surface area and porosity analyzer (Micrometrics ASAP 2020). FTIR spectra were obtained on a FTIR spectrometer (Nicolet 6700, Thermo Electron Scientific Instruments) to record the vibration or stretch of the functional groups in the gel electrolyte. The stretchable properties were investigated using Q800 DMA (TA Instruments). The freezing point of the PAM organohydrogel electrolyte was characterized by a DSC1 (Mettler Toledo) under the $N_2$ atmosphere with a cooling rate of 3 °C min$^{-1}$. The ion conductivity was determined by A.C. Impedance analyzer (E4990A, USA) on a four-electrode cell.

**XAFS measurements and analysis details**. The X-ray absorption fine structure spectra (Fe K-edge) were collected at 1W1B station in Beijing Synchrotron Radiation Facility (BSRF). The storage rings of BSRF were operated at 2.5 GeV with an average current of 250 mA. Using Si(111) double-crystal monochromator, the data collection was carried out in transmission/fluorescence mode using ionization chamber. All spectra were collected in ambient conditions. The acquired EXAFS data were processed according to the standard procedures using the ATHENA module implemented in the IFEFFIT software packages. The k$^3$-weighted EXAFS spectra were obtained by subtracting the post-edge background from the overall absorption and then normalizing with respect to the edge-jump step. Subsequently, k$^3$-weighted χ(k) data of Co K-edge were Fourier transformed to real (R) space using a hanning windows (d$_k$ = 1.0 Å$^{-1}$) to separate the EXAFS contributions from different coordination shells. To obtain the quantitative structural parameters around central atoms, least-squares curve parameter fitting was performed using the ARTEMIS module of IFEFFIT software packages.

**Electrochemical measurement**. The electrochemical tests are carried out in an environmental chamber at 25 °C. The electrochemical measurements were performed in a three-electrode system (CHI 660e workstation, Chenhua, China) in 0.1 M KOH. Saturated calomel electrode (SCE) electrode with saturated 3 M KCl solution and graphite rod acted as reference and counter electrodes, respectively. LSV curves were recorded after 90% IR-compensation. All potentials were converted to the RHE scale according to E$_{(RHE)}$ = E$_{(SCE)}$ + (0.241 + 0.059 pH) V. Rotating ring disk electrode (RRDE-3A, ALS, USA) loaded with catalyst ink (Catalyst loading: 0.3 mg cm$^{-2}$) was used as the working electrode. To prepare the catalyst ink, 6 mg catalysts and 40 μL of Nafion solution were dispersed in 960 μL water-isopropanol solution (volume ratio of 3:1) by sonicating for 1 h to form a homogeneous ink. Then 10 μL of catalyst ink was dropped onto the glass carbon electrode, and dried at 25 °C. The samples were firstly activated in $N_2$-saturated 0.1 M KOH by cyclic voltammetry (CV) method (scan rate of 50 mV s$^{-1}$) to reach a stable condition. The voltage range of CV activation is −1.0 to 0.2 V. When the CV curves completely coincide, the activation process is terminated. Before ORR test, the $O_2$ was bubbled into electrolyte to maintain $O_2$ saturation during the ORR process. LSV curves were recorded in the $O_2$-saturated 0.1 M KOH with a scan rate of 5 mV s$^{-1}$ ranging from 400 to 1600 rpm. The accelerated degradation tests were performed by cycling the catalyst between 0.6 and 1.0 V with a scan rate of 100 mV s$^{-1}$ in $O_2$-saturated 0.1 M KOH. The methanol crossover effect was explored by chronoamperometric responses at 0.70 V. In this process, 5 mL of methanol was injected into the $O_2$-saturated 0.1 M KOH electrolyte. The 4e$^-$ pathway of catalysts was estimated by the following equations:

$$\frac{1}{j} = \frac{1}{j_L} + \frac{1}{j_k} = \frac{1}{B\omega^{1/2}} + \frac{1}{J_k} \tag{1}$$

$$B = 0.2nFC_0 (D_0)^{2/3} \upsilon^{-1/6} \tag{2}$$

$$j_k = nFkC_0 \tag{3}$$

Where $j$ is the measured current density; $j_k$ and $j_L$ represent the kinetic- and diffusion-limiting current densities, respectively; ω is the disc rotation angular velocity of disk, $n$ represents calculated number of the transferred electron in ORR; $F$ is the Faraday constant ($F$ = 96485 C mol$^{-1}$); $C_0$ is the bulk concentration for $O_2$ (1.2 × 10$^{-6}$ mol cm$^{-3}$) dissolved in 0.1 M KOH solution; $D_0$ is the diffusivity of $O_2$ (1.9 × 10$^{-5}$ cm$^2$ s$^{-1}$); $v$ is the kinematic viscosity of electrolyte, and $k$ is the electron-transferred rate constant.

For OER experiments, LSV curves were obtained at a scan rate of 5 mV s$^{-1}$. The $C_{dl}$ was determined by CV measured at different scan rates in the non-faradaic

region. The capacitive current measured at 1.10 V was plotted as a function of scan rate. EIS was measured in the frequency range from 10$^5$ to 0.001 Hz at 1.55 V.

**Aqueous ZABs measurement**. Aqueous ZABs were assembled using the flow cell configuration (20 mL min$^{-1}$ of electrolyte flow rate), not only diluting the byproducts accumulation but also prolonging the stability of batteries. The $O_2$ used comes from air atmosphere. The cycling experiments are carried out in an environmental chamber at 25 °C and 101325 Pa. The design of air cathode adopted a sandwich-like structure, which is composed of catalyst layer (Catalyst loading: 5 mg cm$^{-2}$), water-proof breathable membrane and Ni foam layer. The catalyst layer was prepared by physically mixing the Co SA-NDGs catalyst (60 mg) and acetylene black (10 mg) in 3 mL of ethanol solution, then adding the polytetrafluoroethylene (PTFE) emulsion (60 wt%, 40 μL). After mixing for 30 min and drying at 25 °C for 24 h to remove excessive ethanol, the catalyst layer was cut into 1.0 cm × 1.0 cm pieces. Then, catalyst layer, water-proof breathable membrane and acid-pretreated Ni foam were compressed by roller press, to obtain the air cathode. The air cathode should be kept in vacuum condition before use. A Zn plate (purity 99.9 wt%, 0.3 mm of thickness) was polished with commercial sandpaper for 2 min to remove the oxide layer on the surface. Then the polished Zn plate was used as the anode. For rechargeable ZABs, 6 M KOH + 0.2 M Zn(Ac)$_2$·6H$_2$O was used as electrolyte. For comparison, the aqueous ZABs were also constructed with the commercial Pt/C or Pt/C+RuO$_2$ catalyst as air cathode. The Pt/C or Pt/C+RuO$_2$ catalyst layers were prepared by the same procedure as that used for synthesis of Co SA-NDGs catalyst layer except with 60 mg of Pt/C or 60 mg of Pt/C+RuO$_2$. The galvanostatic discharge curves were recorded by LSVs (scan rate of 5 mV s$^{-1}$) without IR-compensation. Both the current density and power density were normalized to the effective surface area (1 cm$^2$) of the air electrode. The specific capacity and energy density were calculated according to the equation below:

$$\text{Specific capacity} = \frac{\text{current} \times \text{service hour}}{\text{weight of consumed Zn}} \tag{4}$$

$$\text{Energy density} = \frac{\text{current} \times \text{service hour} \times \text{average discharge voltage}}{\text{weight of consumed Zn}} \tag{5}$$

The rate performance and charging/discharging cycling were performed in the NEWARE battery testing system. In generally, one charging/discharging cycle time of ZABs measurement is 22 min unless otherwise specified. To further assess practical applications of aqueous ZABs, one charging/discharging cycle time of aqueous ZABs measurement at 5 mA cm$^{-2}$ is 1 h. One charging/discharging cycle time of symmetric Zn||Zn cell is 10 min.

**Quasi-solid-state ZABs measurement**. The battery experiments are carried out in an environmental chamber at 25, −40, and −60 °C. The design of quasi-solid-state ZABs was typical sandwich-like structure. The $O_2$ used comes from air atmosphere. The quasi-solid-state ZABs measurement was conducted at ~ 25 °C and 101325 Pa. The as-prepared air electrode and Zn plate were placed on the two sides of the PAM gel electrolyte (thickness of 0.5 cm; dimensions of 2 cm × 2 cm × 0.5 cm; mass of 4 g). To maintain good interfacial contact during batteries measurement, the stack pressure (1 kPa) was employed. The extreme low-temperature ZABs measurements were conducted in the low-temperature test chamber (Haier, DW-60W151EU1).

When comparing to existing literature, we performed literature searches of relevant work published in the last five years. By combining the charging/discharging current density and cycling time, the relevant scatter diagram is obtained. Generally, all electrochemical data were measured at least three times. All experimental electrochemical values reported in this manuscript were taken from single experiments.

**Quantum chemistry calculations**. Our first principles calculations were based on density function theory (DFT) using the Perdew-Burke -Ernzerhof (PBE) form for the generalized gradient approximation as implemented in the Vienna ab initio simulation package code[56–58]. The projector augmented wave method with a plane-wave basis set was adopted and the energy cutoff was set to 500 eV. A Monkhorst-centered 7 × 7 × 1 k-mesh was used for structure optimization and total energy calculations. We set the vacuum space to 15 Å along the z-direction, as well as the convergence criteria for total energy and force were set to 10$^{-6}$ eV and 0.01 eV/Å, respectively.

The van der Waals force was involved through the Grimme-D2 scheme to correctly address the adsorption of the molecules and radicals on the surface[59]. The model was built with a CoN$_4$ doped graphene on top of a C$_{60}$ sphere to construct a structure with curvature, and then relaxed for both the volume and the atomic geometries. Then a strain was applied in $xy$ plane and relaxed the atomic positions for a model with larger curvature. A 20 Å vacuum was added in the $z$ direction to exclude the possible interactions due to the periodicity. The cutoff energy of the plane wave basis set was taken as 450 eV, and only the gamma point was used for the calculations. The convergence thresholds of energy and force were set to 0.01 eV and 10$^{-6}$ eV Å$^{-1}$, respectively. Moreover, the Coulomb interaction strengths (U and J parameters) were self-consistently achieved with a local screened

Coulomb correction (LSCC) approach[60], and the interactions were incorporated through a DFT + U scheme[61].

**Adsorption energy calculations.** In adsorption energy calculations, Density functional theory (DFT) calculations were performed using the Vienna ab initio simulation package (VASP), based on projector-augmented-wave (PAW) potential. The Perdew-Burke-Ernzeralized gradient approximation (GGA) using the approach of Grimme (DFT-D3) is used to describe the van der Waals force (vdW) between atoms. All structural models adopt cut-off energy of 500 eV, and choose Monkhorst-Pack method as the Brillouin zone integral calculation scheme. The convergence criterion for self-consistent field iteration is $1 \times 10^{-6}$ eV; the convergence criterion for structural optimization is 0.01 eV/Å. The adsorption energy $E_{ads}$ can be calculated by the following equation:

$$E_{ads} = E_{total} - E_{polymer} - E_{water} \qquad (6)$$

where $E_{total}$ is the total energy, $E_{polymer}$ is the energy of the polymer, and $E_{water}$ is the energy of the water. The higher absolute value of $E$ indicates stronger interactions.

## Data availability

The data that support the findings of this study are available from the corresponding author upon reasonable request. Source data are provided with this paper.

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

## Acknowledgements

This work was financially supported by the Hunan Provincial Science and Technology Plan Project (2020JJ4710). Feng is supported by the Research start-up Funds (Grant No.~2019KY23) from Southwest Jiaotong University. We sincerely thank Prof. Dingsheng Wang and Prof. Yadong Li for their valuable suggestions.

## Author contributions

Q.W. and Q.F. contributed equally to this work. Q.W., Y.W., P.Y., G.F. and J. L. performed the catalyst synthesis, characterizations, and electrochemical experiments, collected and analyzed the data. W.L. performed the XAFS measurement. Q.F., S.T. and L.X. conducted and discussed the theoretical calculations and molecular dynamics simulations. X. X. supervised the research. Y. L. designed the project and coordinated all research work and wrote the manuscript. All authors discussed the result, and contributed to the writing of the manuscript.

## Competing interests

The authors declare no competing interests.
