## [Peer Review File · Nature Communications]

Reviewer comments, first round –

Reviewer #1 (Remarks to the Author):

In this work, the author reported an ultrahigh-rate and robust quasi-solid-state ZABs integrated with the atomic Co-N₄ sites anchored on wrinkled nitrogen-doped graphene (Co SA-NDGs) cathode and the modulated H-bond network of polyacrylamide (PAM) organohydrogel electrolyte with DMSO. It is quite impressive to see high rate performance in such a wide temperature range. However, some additional data still need to be acquired before it can be suitable for publication in Nature Communications.

- 1)The authors showed C, N, Co XPS data, how about oxygen 1s spectrum? Are there any interactions between Co and O? From the survey XPS shown in Figure S8, the oxygen percentage seems high.
- 2)Can the Co-N₄ structure explain the mixed Co²⁺ and Co³⁺ oxidation states noted in the Co 2p XPS data in the supporting information? What are the corresponding Co and N oxidation state in the Co-N₄ structure unit? Please specify.
- 3)The authors performed detailed in situ and ex situ characterizations of Zn anodes before and after cycling in Zn-air batteries, however, some post chem characterizations of the cathodes will be necessary to demonstrate the structural stability and integrity of the cathodes.
- 4)Although the authors demonstrated the strong H-bonding capability of DMSO, the toxicity of this solvent should not be overlooked. Are there safer alternatives available that still serve similar purpose?

Some typos in the text and figures need to corrected:

- 1)Abstract, line 54"Here we present.. organohydrogel electrolyte with.." Clearly, the sentence is not completed.
- 2) Supporting information Figure 15a, x axis title should be potential instead of current density.

Reviewer #2 (Remarks to the Author):

In this paper, the authors developed a quasi-solid-state Zinc-Air-Batteries (ZAB) with Co-N₄ sites supported on the interconnected porous nitrogen-graphene. Also, the electrolyte is synthesized by dimethyl sulfoxide into polyacrylamide to enhance interface stability and adaptability for a large range of working temperatures. The ZAB can perform high-rate electrochemical performance in various conditions.

1. Some papers like Single Co atoms anchored in porous N-doped carbon for efficient zinc– air battery cathodes (ACS Catalysis, 2018, 8: 8961-8969.) and Atomic cobalt on defective bimodal mesoporous carbon toward efficient oxygen reduction for zinc–air batteries (Small Methods, 2019, 3: 1800450.) have reported the single Co atoms in N-doped carbon in ZABs. The methods are similar to this paper, thus, this manuscript is not of sufficient novelty.
2. In rate performance and discharge-charge cycling, the test time (10 min) for each step is too short which is not enough for practical applications. The cycling performance of long discharge/charge time (1-4 hours) for each cycle should be added.
3. One key highlight of this manuscript is the adaptability of hydrogel electrolytes in a large range of working temperatures. Therefore, the retention of H₂O in organ-hydrogel electrolytes, under different temperatures, as an important characterization, should be analyzed.
4. In Figure 4, charging/discharging cycling performance at a low current density at -60 °C was displayed. How about cycling performance at high current density (50 mA cm⁻² or 100 mA cm⁻²) at -60 °C?

Reviewer #3 (Remarks to the Author):

The authors reported an interesting quasi-solid-state Zn-air battery with a high cycling rate and

high-capacity retention. The capacity retention is remarkable even under $-60\text{ }^{\circ}\text{C}$, due to the formation of hydrogen bonds between the solvent DMSO and PAM hydrogel. Both experimental and theoretical work had been carried out to characterize the performance of the battery. The results are interesting, but the manuscript should be revised and checked carefully since there are quite a few grammatical mistakes. Here I list a few of them.

1. In the abstract, the sentence "the modulated H-bond network of polyacrylamide (PAM) organohydrogel electrolyte with" should be checked.
2. On page 11: "Density of states (DOS) analysis (Fig. 2h) show that" and "charge transfer difference".
3. On page 14: "reconstructing the solvation Zn^{2+} solvation sheath structure".
4. On page 15: "ZABs worked below $-40\text{ }^{\circ}\text{C}$ is seldomly reported".
5. On page 22: the details of the dynamics in "Molecular dynamics simulations" is missing. There were no dynamical results presented in the manuscript. If only optimization had been performed in the computational part, this section should be merged into the following section "Quantum chemistry calculations".
6. The full form should be used when you mentioned the abbreviation for the first time, such as "Vienna ab initio simulation package (VASP)" on page 23.

The manuscript could be published if the authors revised the grammatical mistakes carefully.

Responses to the Referees' Comments

Manuscript ID: NCOMMS-21-46171

Title: Ultrahigh-rate quasi-solid-state Zn-air batteries enabled by atomically dispersed Co site electrocatalyst and organhydrogel electrolyte

We thank the referees for their valuable comments and positive endorsement to our manuscript. We have carefully considered the referees' comments and revised the manuscript accordingly. Our responses and corresponding revisions are as follows:

Reviewer #1 (Remarks to the Author):

In this work, the author reported an ultrahigh-rate and robust quasi-solid-state ZABs integrated with the atomic Co-N₄ sites anchored on wrinkled nitrogen-doped graphene (Co SA-NDGs) cathode and the modulated H-bond network of polyacrylamide (PAM) organohydrogel electrolyte with DMSO. It is quite impressive to see high-rate performance in such a wide temperature range. However, some additional data still need to be acquired before it can be suitable for publication in Nature Communications.

1)The authors showed C, N, Co XPS data, how about oxygen 1s spectra? Are there any interactions between Co and O? From the survey XPS shown in Figure S8, the oxygen percentage seems high.

Response: Thank you very much for your professional advice. In generally, the O element mainly comes from oxygen-containing functional groups and absorbed H₂O molecule. Although the most oxygen-containing groups after high-temperature treatment are mostly removed, it can't be eliminated. **From the fitted O 1s spectra (Figure R1), the three main peaks are assigned to C=O, C-O-C and C-OH. These hydrophilic oxygen-containing-groups enhance the three-phase contact.**^[1] Similar phenomena have also appeared in other references [*Nat. Commun.* **2020**, *11*, 3049].^[2]

The relevant content has been added to the revised Manuscript and Supplementary Information (Supplementary Figure 8c), which were marked in yellow.

Figure R1. O 1s spectra for Co SA-NDGs.

2) Can the Co-N₄ structure explain the mixed Co²⁺ and Co³⁺ oxidation states noted in the Co 2p XPS data in the supporting information? What are the corresponding Co and N oxidation state in the Co-N₄ structure unit? Please specify.

Response: Thank you very much for your professional advice. XPS is a surface analysis tool. Firstly, we compared the peak splitting of Co/Fe/Cu 2p in other reported SAC catalysts (**Figure R2**).^[2-8] The 2p_{3/2} orbit, 2p_{1/2} orbit and satellite peaks are both observed. Owing to the trace content of Co (0.14 wt.% measured by ICP-OES), there are inevitable errors in distinguishing the valence. From the Co 2p XPS data of Co SA-DNGs, the main signals located at the binding energies of 780.1 and 795.7 eV are attributed to the Co²⁺, and the peaks centered at the binding energies of 782.1 and 797.6 eV are characteristic of the Co³⁺. **The peak intensity of Co²⁺ is obviously higher than that of Co³⁺, demonstrating the main oxidation state of Co species is +2.**

The average result of oxidation state is obtained from the corresponding Co K-edge of Co SA-NDGs in X-ray absorption near-edge structure [*Nano Lett.* **2020**, *20*, 5443].^[9] Compared with the reference samples, **the average oxidation state of Co species in Co SA-NDGs is about +2.19 (Figure R3).**

Figure R2. Comparison the peak splitting of Co/Fe/Cu 2p in other reported SAC catalysts.

Figure R3. Co K-edge XANES of Co foil, CoO, Co₃O₄, and Co SA-NDGs. (b) The fitted oxidation states of Co.

Besides, we have performed a Bader charge analysis for our calculations. It reveals that, when there is no molecules or radicals adsorbed, the Co atom will lose about 1.1 electrons for both the modeled planar and curved structures, while the neighboring N atom will gain about 2.62 electrons. For curved structure, the Co atom will lose more electrons and N atom will gain more electron comparing to those in the planar structure, respectively. When OH was adsorbed, Co atom will be more oxidized. The detailed data is supplied as **Table R1**. Similar phenomena are observed when another radical is adsorbed on the active site.

According to the literature [*Nat. Catal.* **2021**, *4*, 407],^[10] the calculation Bader charge is approximately 50% compared with the fact. Therefore, the calculation result is consistent with the XANES result. **In the Co-N₄ structure, the corresponding Co and N oxidation state is $\sim +2.2$ and -5.5 , respectively.**

Table R1. Lose or gain of electrons for each atom in Co-N₄ from Bader charge analysis

Atoms	No adsorption		OH adsorption	
	Planar Co-N ₄	Curved Co-N ₄	Planar Co-N ₄	Curved Co-N ₄
Co	-1.08395	-1.10007	-1.43799	-1.31096
N1	2.574933	2.675851	2.558905	2.782011
N2	2.574935	2.676346	2.558941	2.78261
N3	2.574933	2.675851	2.558939	2.78201
N4	2.574935	2.676346	2.558941	2.78261

The relevant content has been added to the revised Manuscript and Supplementary Information (Supplementary Figure 9b and Table 3), which were marked in yellow.

3) The authors performed detailed in situ and ex situ characterizations of Zn anodes before and after cycling in Zn-air batteries, however, some post echem characterizations of the cathodes will be necessary to demonstrate the structural stability and integrity of the cathodes.

Response: Thank you very much for your professional advice. According to your valuable suggestion, we have added the related characterization (e.g., SEM, TEM, and XPS) of air cathode after cycling. **As seen in Figure R4, the air cathode still retains a 3D interconnected porous architecture with a highly wrinkled surface. TEM image shows that the ultrathin graphene nanosheet is highly curved. From XPS analysis (Figure R5), the atomic Co-N₄ sites is well-maintained.** These results sufficiently demonstrate the structural stability and integrity of the cathodes during the long-term cycling.

Figure R4. (a) SEM image, (b) TEM image for Co SA-NDGs after the long-term cycling.

Figure R5. (a) XPS survey spectra, (b) Co 2p spectra, (c) N 1s spectra, (d) C 1s

spectra and (e) O 1s spectra for Co SA-NDGs after the long-term cycling.

The relevant content has been added to the revised Manuscript and Supplementary Information (Supplementary Figure 29 and 30), which were marked in yellow.

4) Although the authors demonstrated the strong H-bonding capability of DMSO, the toxicity of this solvent should not be overlooked. Are there safer alternatives available that still serve similar purpose?

Response: Thank you very much for your professional advice. In this work, the dimethyl sulfoxide (DMSO) plays a role of electrolyte additive adhere to the characteristics of “small dosage” and “quick effect”. **Table R2** shows the physical properties of commonly used organic solvents (e.g. dimethyl sulfoxide, ethyl acetate, ethylene carbonate, dichloromethane, dimethyle carbonate, ethyl methyl carbonate, diethyl carbonate, acetonitrile, propylene carbonate, dimethyl formamide, tetrahydrofuran, ethylene glycol, glycerol, isopropyl alcohol, ethanol and methanol) for energy storage systems.^[11-19] The following factors should be considered in the selection of electrolyte additives, such as viscosity, freezing point, boiling point, flash point, density and toxicity. On the one hand, **DMSO is a sulfur-containing compound with strong polarity, a high boiling point, low toxicity, and good chemical stability. DMSO is a common organic solvent with good solubility with most inorganic and organic compounds, especially when dissolved with water in any ratio.** In DMSO/H₂O solution, the strong H-bond interactions between water and DMSO molecules significantly weaken the H-bond interactions within the water molecules, and the freezing point of the binary solution system decreases significantly. Additionally, the strong inter-molecular H-bonds between DMSO and H₂O result in a decrease in the saturated vapor pressure of H₂O molecular, preventing the evaporation of H₂O at elevated temperatures. On the other hand, **DMSO is stable to alkali.** Importantly, **its toxicity is lower than other solvents.** Among these, DMSO is the best choice.

Besides, **high-concentration electrolyte and quasi-solid-state/solid-state electrolyte with tunable functional groups can achieve the same purpose, but the final effect is slightly different.**^[20,21] In detail, the strong dipole-dipole force between Zn^{2+} and O rearranges the H_2O coordination structure in concentrated electrolyte, in which the O is more bound by metal ions instead of forming H-bond.^[22] Taking the steric hindrance in consideration, the larger anions with low charge density can also weaken the solvation effect.^[23] Contrary to “salt-in-water” aqueous electrolyte, the “water-in-salt” solution, which is like the ionic liquid, was proposed with surprisingly high salt concentration and low water concentration.^[24] From the perspective of binding H-bonds of H_2O molecules, quasi-solid gel electrolyte or even solid electrolyte is the most effective.^[25-27] Hydrogels are usually composed of flexible crosslinked hydrated polymer chains, which possess rich hydrophilic functional groups such as -OH, -COOH, - SO_3 , - NH_2 to form intramolecular or intermolecular H-bonds. In principle, the competition between H_2O molecules and different chemical functional groups controls the antifreeze performance of hydrogel electrolyte.

Table R2. Physical properties of commonly used organic solvents for energy storage systems.

Organic solvents	CAS number	Viscosity (mPa s)	Freezing point (°C)	Boiling point (°C)	Flash point (°C)	Density (g cm⁻³)	Toxicity
Dimethyl sulfoxide	67-68-5	2.00	19	189	89	1.10	Low toxicity
Ethyl acetate	141-78-6	0.45	-84	77	-4	0.90	Low toxicity
Ethylene carbonate	96-49-1	1.90	34-37	243	150	1.32	Low toxicity
Dichloromethane	75-09-2	0.43	-95	40		1.33	Moderate toxicity
Dimethyle carbonate	616-38-6	0.63	3	90	17	1.06	Low toxicity
Ethyl methyl carbonate	623-53-0	0.65	-14	107	23	1.01	High toxicity
Diethyl carbonate	105-58-8	0.75	-43	127	25	0.97	Moderate toxicity
Acetonitrile	75-05-8	0.34	-44	81.6	5.6	0.78	Moderate toxicity
Propylene carbonate	108-32-7	2.53	-48.8	242	132	1.20	Low toxicity
Dimethyl formamide	68-12-2	0.94	-60.5	153	58	0.94	High toxicity
Tetrahydrofuran	109-99-9	0.53	-108.5	66	17	0.89	High toxicity
Ethylene glycol	107-21-1	25.66	-12.9	197.3	111	1.113	Moderate toxicity
Glycerol	56-81-5	800	18.17	290	176	1.261	Low toxicity
Isopropyl alcohol	67-63-0	1.17	-89.5	82.5	11.7	0.785	High toxicity
Ethanol	64-17-5	1.18	-114.1	78.3	12	0.789	Low toxicity
Methanol	67-56-1	0.80	-97.8	64.8	11.11	0.791	High toxicity

Some typos in the text and figures need to be corrected:

1) Abstract, line 54 "Here we present.. organohydrogel electrolyte with.." Clearly, the sentence is not completed.

Response: Thank you very much for your professional advice. This is our mistake.

We have now corrected this mistake. Therefore, the complete sentence is as follows:

Here we present an ultrahigh-rate and robust quasi-solid-state ZABs integrated with the atomic Co-N₄ sites anchored on wrinkled nitrogen-doped graphene (Co SA-NDGs) cathode and the polyacrylamide (PAM) organohydrogel electrolyte with the modulated H-bond network.

2) Supporting information Figure 15a, x axis title should be potential instead of current density.

Response: Thank you very much for your professional advice. According to your valuable suggestion, we have changed the x axis title of into Potential (V vs. RHE) in

Figure R6a.

Figure R6. LSV curves of (a) Co SA-NDGs and (b) NDGs measured at different temperature.

The relevant mistake has been corrected in the revised Supplementary Information (Supplementary Figure 15a), which were marked in yellow.

Reference

1. Q. Wang, Y. Lei, Z. Chen, N. Wu, Y. Wang, B. Wang, Y. Wang, Fe/Fe₃C@C nanoparticles encapsulated in N-doped graphene-CNTs framework as an efficient bifunctional oxygen electrocatalyst for robust rechargeable Zn-air batteries. *J. Mater. Chem. A* **6**, 516-526 (2018).
2. H. Shang, X. Zhou, J. Dong, A. Li, X. Zhao, Q. Liu, Y. Lin, J. Pei, Z. Li, Z. Jiang, D. Zhou, L. Zheng, Y. Wang, J. Zhou, Z. Yang, R. Cao, R. Sarangi, T. Sun, X. Yang, X. Zheng, W. Yan, Z. Zhuang, J. Li, W. Chen, D. Wang, J. Zhang, Y. Li, Engineering unsymmetrically coordinated Cu-S₁N₃ single atom sites with enhanced oxygen reduction activity. *Nat. Commun.* **11**, 3049-3059 (2020).
3. Y. Pan, S. Liu, K. Sun, X. Chen, B. Wang, K. Wu, X. Cao, W. Cheong, R. Shen, A. Han, Z. Chen, L. Zheng, J. Luo, Y. Lin, Y. Liu, D. Wang, Q. Peng, Q. Zhang, C. Chen, Y. Li, A bimetallic Zn/Fe polyphthalocyanine-derived single-atom Fe-N₄ catalytic site: a superior trifunctional catalyst for overall water splitting and Zn-air batteries. *Angew. Chem. Int. Ed.* **57**, 8614-8618 (2018).
4. Y. He, S. Hwang, D. A. Cullen, M. A. Uddin, L. Langhorst, B. Li, S. Karakalos, A. J. Kropf, E. C. Wegener, J. Sokolowski, M. Chen, D. Myers, D. Su, K. L. G. Wang, S. Litster, G. Wu, Highly active atomically dispersed CoN₄ fuel cell cathode catalysts derived from surfactant-assisted MOFs: carbon-shell confinement strategy. *Energy Environ. Sci.* **12**, 250-260 (2019).
5. K. Chen, S. Kim, M. Je, H. Choi, Z. Shi, N. Vladimir, K. H. Kim, O. L. Li, Ultrasonic plasma engineering toward facile synthesis of single-atom M-N₄/N-doped carbon (M=Fe, Co) as superior oxygen electrocatalyst in rechargeable zinc-air batteries. *Nano-Micro Lett.* **13**, 60-79 (2021).
6. J. Yang, Z. Wang, C. X. Huang, Y. Zhang, Q. Zhang, C. Chen, J. Du, X. Zhou, Y. Zhang, H. Zhou, L. Wang, X. Zheng, L. Gu, L. M. Yang, Y. Wu, Compressive strain modulation of single iron sites on helical carbon support boosts electrocatalytic oxygen reduction. *Angew. Chem. Int. Ed.* **60**, 22722-22728 (2021).
7. H. Zhang, S. Hwang, M. Wang, Z. Feng, S. Karakalos, L. Luo, Z. Qiao, X. Xie, C.

- Wang, D. Su, Y. Shao, G. Wu, Single atomic iron catalysts for oxygen reduction in acidic media: particle size control and thermal activation. *J. Am. Chem. Soc.* **139**, 14143-14149 (2017).
8. W. Liu, L. Zhang, X. Liu, X. Liu, X. Yang, S. Miao, W. Wang, A. Wang, T. Zhang, Discriminating catalytically active FeN_x species of atomically dispersed Fe-N-C catalyst for selective oxidation of the C-H bond. *J. Am. Chem. Soc.* **139**, 10790-10798 (2017).
9. H. Shang, W. Sun, R. Sui, J. Pei, L. Zheng, J. Dong, Z. Jiang, D. Zhou, Z. Zhuang, W. Chen, J. Zhang, D. Wang, Y. Li, Engineering isolated Mn-N₂C₂ atomic interface sites for efficient bifunctional oxygen reduction and evolution reaction. *Nano Lett.* **20**, 5443-5450 (2020).
10. S. Ji, B. Jiang, H. Hao, Y. Chen, J. Dong, Y. Mao, Z. Zhang, R. Gao, W. Chen, R. Zhang, Q. Liang, H. Li, S. Liu, Y. Wang, Q. Zhang, L. Gu, D. Duan, M. Liang, D. Wang, X. Yan, Y. Li, Matching the kinetics of natural enzymes with a single-atom iron nanozyme. *Nat. Catal.* **4**, 407-417 (2021).
11. X. Dong, Y. Lin, P. Li, Y. Ma, J. Huang, D. Bin, Y. Wang, Y. Qi, Y. Xia, High-energy rechargeable metallic lithium battery at -70 °C enabled by a cosolvent electrolyte. *Angew. Chem. Int. Ed.* **58**, 5623-5627 (2019).
12. Q. Rong, W. Lei, J. Huang, M. Liu, Low temperature tolerant organohydrogel electrolytes for flexible solid-state supercapacitors. *Adv. Energy Mater.* **8**, 1801967 (2018).
13. J. Chen, J. Vatamanu, L. Xing, O. Borodin, H. Chen, X. Guan, X. Liu, K. Xu, W. Li, Improving electrochemical stability and low-temperature performance with water/acetonitrile hybrid electrolytes. *Adv. Energy Mater.* **10**, 1902654 (2019).
14. A. Naveed, H. Yang, Y. Shao, J. Yang, N. Yanna, J. Liu, S. Shi, L. Zhang, A. Ye, B. He, J. Wang. A highly reversible Zn anode with intrinsically safe organic electrolyte for long-cycle-life batteries. *Adv. Mater.* **31**, 1900668 (2019).
15. N. Chang, T. Li, R. Li, S. Wang, Y. Yin, H. Zhang, X. Li, An aqueous hybrid electrolyte for low-temperature zinc-based energy storage devices. *Energy Environ.*

- Sci.* **13**, 3527-3535 (2020).
16. J. Holoubek, H. Liu, Z. Wu, Y. Yin, X. Xing, G. Cai, S. Yu, H. Zhou, T. A. Pascal, Z. Chen, P. Liu. Tailoring electrolyte solvation for Li metal batteries cycled at ultra-low temperature. *Nat. Energy* **6**, 303-313 (2021).
 17. R. Qin, Y. Wang, M. Zhang, Y. Wang, S. Ding, A. Song, H. Yi, L. Yang, Y. Song, Y. Cui, J. Liu, Z. Wang, S. Li, Q. Zhao, F. Pan, Tuning Zn²⁺ coordination environment to suppress dendrite formation for high-performance Zn-ion batteries. *Nano Energy* **80**, 105478-105486 (2021).
 18. C. Lu, X. Chen, All-temperature flexible supercapacitors enabled by antifreezing and thermally stable hydrogel electrolyte. *Nano Lett.* **20**, 1907-1914 (2020).
 19. D. Han, C. Cui, K. Zhang, Z. Wang, J. Gao, Y. Guo, Z. Zhang, S. Wu, L. Yin, Z. Weng, F. Kang, Q. H. Yang, A non-flammable hydrous organic electrolyte for sustainable zinc batteries. *Nat. Sustainab.* DOI: 10.1038/s41893-021-00800-9 (2021).
 20. Z. Liua, X. Luo, L. Qin, G. Fang, S. Liang, Progress and prospect of low-temperature zinc metal batteries, *Adv. Powder Mater.* DOI:10.1016/j.apmate.2021.10.004, (2021).
 21. Q. Nian, T. Sun, S. Liu, Haihui Du, X. Ren, Z. Tao, Issues and opportunities on low-temperature aqueous batteries, *Chem. Eng. J.* **423**, 130253-130262 (2021).
 22. Q. Zhang, Y. Ma, Y. Lu, L. Li, F. Wan, K. Zhang, J. Chen, Modulating electrolyte structure for ultralow temperature aqueous zinc batteries, *Nat. Commun.* **11**, 4463-4472 (2020).
 23. N. Zhang, F. Cheng, Y. Liu, Q. Zhao, K. Lei, C. Chen, X. Liu, J. Chen, Cation-deficient spinel ZnMn₂O₄ cathode in Zn(CF₃SO₃)₂ electrolyte for rechargeable aqueous Zn-ion battery, *J. Am. Chem. Soc.* **138**, 12894-12901 (2016).
 24. L. Ma, N. Li, C. Long, B. Dong, D. Fang, Z. Liu, Y. Zhao, X. Li, J. Fan, S. Chen, S. Zhang, C. Zhi, Achieving both high voltage and high capacity in aqueous zinc ion battery for record high energy density, *Adv. Funct. Mater.* **29**, 1906142 (2019).
 25. H. Zhang, X. Liu, H. Li, B. Qin, S. Passerini, High-voltage operation of a V₂O₅

- cathode in a concentrated gel polymer electrolyte for high-energy aqueous zinc batteries, *ACS Appl. Mater. Interfaces* **12**, 15305-15312 (2020).
26. X. Li, L. Ma, Y. Zhao, Q. Yang, D. Wang, Z. Huang, G. Liang, F. Mo, Z. Liu, C. Zhi, Hydrated hybrid vanadium oxide nanowires as the superior cathode for aqueous Zn battery, *Mater. Today Energy* **14**, 100361-100367 (2019).
27. Q. Yang, F. Mo, Z. Liu, L. Ma, X. Li, D. Fang, S. Chen, S. Zhang, C. Zhi, Activating C-coordinated iron of iron hexacyanoferrate for Zn hybrid-ion batteries with 10000-cycle lifespan and superior rate capability, *Adv. Mater* **31**, 1901521 (2019).

Finally, we sincerely thank you for your valuable and sincere suggestions. According to your suggestions, we believe that the revised manuscript will be more scientific and rigorous

Reviewer #2 (Remarks to the Author):

In this paper, the authors developed a quasi-solid-state zinc-air-batteries (ZABs) with Co-N₄ sites supported on the interconnected porous nitrogen-graphene. Also, the electrolyte is synthesized by dimethyl sulfoxide into polyacrylamide to enhance interface stability and adaptability for a large range of working temperatures. The ZAB can perform high-rate electrochemical performance in various conditions.

1. Some papers like Single Co atoms anchored in porous N-doped carbon for efficient zinc-air battery cathodes (ACS Catalysis, 2018, 8: 8961-8969.) and Atomic cobalt on defective bimodal mesoporous carbon toward efficient oxygen reduction for zinc-air batteries (Small Methods, 2019, 3: 1800450.) have reported the single Co atoms in N-doped carbon in ZABs. The methods are similar to this paper, thus, this manuscript is not of sufficient novelty.

Response: Thank you very much for your professional advice. Over the past decade, a variety of atomically dispersed M-N-C catalysts (M=Fe/Co/Mn/Cu, etc) have been investigated for ORR and zinc-air battery. We carefully compared the differences between the two published papers and our manuscript in terms of synthesis, active site, electrocatalytic activity, and battery performance (Table R3). **Although they are all Co single-atom catalysts, their synthetic methods and active sites are different, which also leads to different performance. In our manuscript, we achieve a significant breakthrough in assembled quasi-solid-state ZABs' performance by atomically dispersed Co site electrocatalyst and organhydrogel electrolyte. We focus on the important role of graphene curvature, interface stability and ultra-low-temperature performance of quasi-solid-state ZABs.** Next, we will list the highlights and distinctive research content of this work.

Table R3. Literature comparison with our work.

Catalysts	Synthesis	Performance	Active sites	Aqueous ZABs	Quasi-solid-state ZABs	Reference
Co SA-NDGs (atomic Co-N ₄ sites anchored on nitrogen-doped graphenes)	Hydrothermal method, annealing and acid leaching	ORR: E _{onset} : 1.02 V E _{1/2} : 0.87 V OER: E _{j=10} : 1.58 V	Curved Co-N ₄ site	P_{max} : 251.4 mW cm ⁻² Specific capacity: 757.4 mAh g ⁻¹ Cycling: 300 h@10 mA cm ⁻²	P_{max} : 219.9 mW cm ⁻² Cycling: 100 h@50 mA cm ⁻² 50 h@100 mA cm ⁻²	This work
NC-Co SA (Co single atoms anchored in porous N-doped carbon nanoflake arrays)	Co-MOF annealing and acid leaching	ORR: E _{onset} : 1.00 V E _{1/2} : 0.87 V OER: E _{j=10} : 1.59 V	Co-N _x site	Cycling: 180 h	P_{max} : 20.9 mW cm ⁻² Cycling: 41 h	ACS Catal. 2018 , 8, 8961.
A-Co@CMK-3-D (Co single atoms supported on defective bimodal mesoporous carbon)	Hard-template method, annealing and acid leaching	ORR: E _{onset} : 0.946 V E _{1/2} : 0.835 V	Co-C site	P_{max} : 162 mW cm ⁻² Specific capacity: 765 mAh g ⁻¹ Cycling: 45 h@10 mA cm ⁻²		Small Methods 2019 , 3, 1800450.

Highlights:

1. O₂ reduction/evolution activity of Co-N₄ sites supported on the interconnected porous nitrogen-doped graphenes (Co SA-NDGs) is enhanced by curvature, contributing to decrease charging/discharging polarization and high-rate capability.
2. The introduction of dimethyl sulfoxide into polyacrylamide hydrogel electrolyte effectively modulates the H-bond network, enabling improved interface stability and all-temperature adaptability (from -60 to 60°C) simultaneously.
3. The quasi-solid-state Zn-air batteries with Co SA-NDGs not only achieve the record-high cycling rates of 100 mA cm⁻² over 50 h at room temperature, but also show exceptional cycling stability with capacity retention over 90% more than 300 h (0.5 mA cm⁻²) at -60 °C.

➤ The synthetic process of Co SA-NDGs

Figure R7. (a) Schematic illustration of the fabrication process for Co SA-NDGs. Optical photograph of (b) rGO-based hydrogel and (c) Co SA-NDGs.

The synthetic procedure of Co SA-NDGs was described in **Figure R7**, including electrostatic adsorption, hydrothermal treatment, subsequent pyrolysis and acid leaching.

Different from the Co-MOF pyrolysis and hard-template method, our synthesis method is helpful to make use of the three-dimensional porous structure of N-doped graphene and enhance the mass transfer efficiency at high-rate measurement.

➤ **The evolution process of atomically dispersed Co atoms**

Figure R8. In situ temperature-dependent TEM images recorded at varying temperature.

Figure R9. Molecular dynamic simulations to reveal the dynamic evolution of Co dimer. The pink, blue, and gray balls refer to Co, N, and C atoms, respectively.

To deeply understand the evolution process of atomically dispersed Co atoms during pyrolysis, in situ temperature-dependent TEM investigations and molecular dynamics (MD) simulations were performed. The structural evolution was timely monitored in the temperature window from 25 to 600 °C (Figure R8). In our case, the reduced graphene oxide (rGO) becomes less crystallization as the temperature rises to 600 °C, which may be ascribed to the heteroatom doping effect. More interestingly, the related Co-based nanoparticles or clusters do not appear. Furthermore, based on our previous model, we place Co dimers on tetrapyrroline N-doped carbon sites that act synergistically with vacancy defects to examine possible changes at 800 °C. The results show that two Co single atoms do not form a stable Co dimer (Figure R9). The Co-Co bond of the dimer gradually becomes longer until it is broken. The Co connected to the tetrapyrroline N automatically forms a typical planar Co-N₄ configuration, while other Co atom will become more and more away from the Co-N₄ coordination. This result fully proves that stable Co dimers are not easy to form, but due to the synergistic effect of charge transfer and vacancy defects, the typical planar four-coordinate configuration of Co-N₄ is formed. Simultaneously, the escaped Co atom may be captured by the doped N species or graphene defects until it

is stabilized. These experimental observations and MD simulations provide solid evidences of the evolution process for the atomically dispersed Co-N₄ structure during synthesis.

Compared to the two above-mentioned references, we obtain an insightful understanding of the evolution process of atomically dispersed Co atoms during pyrolysis.

➤ **The bifunctional activity of Co SA-NDGs**

Figure R10. Comparison of bifunctional ORR/OER activities of Co SA-NDGs and other non-precious metal catalysts reported.

Generally, the bifunctional activity is evaluated by ΔE ($\Delta E = E_{\text{OER};j=10} - E_{\text{ORR};1/2}$). We use $E_{1/2}$ and E_{10} index as the horizontal and vertical axis for better comparison, where the point located at the upper right demonstrates a better bifunctional capability. **The ΔE of Co SA-NDGs is only 0.71 V, ranking the top level among non-precious metal SACs and metal nanoparticle-based bifunctional catalysts (Figure R10 and Supplementary Table 2 in the revised Supplementary Information).**

A-Co@CMK-3-D [*Small Methods* **2019**, *3*, 1800450] is ORR catalyst with a $E_{1/2}$ of 0.835 V, and NC-Co SA [*ACS Catal.* **2018**, *8*, 8961.] is bifunctional ORR/OER catalyst with a ΔE of 0.72 V. **Obviously, Co SA-NDGs is the best catalyst among them.** The high activity is enhanced by curvature, contributing to decrease charging/discharging polarization and high-rate capability.

➤ **DFT calculation of curvature on atomic Co-N₄-C system**

Figure R11. (a) Established model of planar Co-N₄ site and curved Co-N₄ site, (b) Calculated differential charge density. Yellow and blue areas represent charge density aggregation and depletion, respectively. (c) Density of states for planar Co-N₄ site and curved Co-N₄ site. (d) Free energy diagram of the ORR on planar Co-N₄ site and curved Co-N₄ site in alkaline media. (e) The optimized intermediate species along the reaction pathway of ORR on curved Co-N₄.

The crucial role of curvature on atomic Co-N₄-C system was studied using

first-principles calculations based on DFT (Figure R11a). Figure R11b gives the calculated differential charge densities for the planar Co-N₄ and curved Co-N₄ model to show the charge transfer difference. The curved Co-N₄ shows more localized charge densities and a larger charge gradient than the planar Co-N₄, which is believed to facilitate the subsequent O₂ activation. Density of states (DOS) analysis (Figure R11c) shows that the d-band center of curved Co-N₄ is downshifted compared to that of planar Co-N₄, indicating that the adsorption of oxygenated intermediates is lowered. We have performed a Bader charge analysis for our calculations. It reveals that, when there is no molecules or radicals adsorbed, the Co atom will lose about 1.1 electrons for both the modeled planar and curved structures, while the neighboring N atom will gain about 2.62 electrons. For curved structure, the Co atom will lose more electrons and N atom will gain more electron comparing to those in the planar structure, respectively. When OH was adsorbed, Co atom will be more oxidized. The detailed data is supplied as Table R4. Similar phenomena are observed when another radical is adsorbed on the active site. Particularly, the ORR activity of bifunctional electrocatalyst determines the energy conversion efficiency to a great extent. The free energy pathways of two models are both downhill on the four-electron ORR processes at $U = 0$ V, revealing that each elementary step can be carried out spontaneously (Figure R11d). At $U = 1.23$ V, the rate-determining step is the first protonation of the adsorbed OOH* species. The ORR overpotential of curved Co-N₄ site under alkaline condition is calculated to be 0.758 eV, which is lower than that of planar Co-N₄ site (0.90 eV), suggesting **the beneficial effect of graphene curvature for optimizing the ORR activity**. Meanwhile, the stable adsorption configuration of intermediates is provided (Figure R11e).

Table R4. Lose or gain of electrons for each atom in Co-N₄ from Bader charge analysis

Atoms	No adsorption		OH adsorption	
	Planar Co-N ₄	Curved Co-N ₄	Planar Co-N ₄	Curved Co-N ₄
Co	-1.08395	-1.10007	-1.43799	-1.31096

N1	2.574933	2.675851	2.558905	2.782011
N2	2.574935	2.676346	2.558941	2.78261
N3	2.574933	2.675851	2.558939	2.78201
N4	2.574935	2.676346	2.558941	2.78261

➤ **Characterizations of organhydrogel electrolyte and interface stability**

Figure R12. (a) Zn^{2+} solvation structure and formed H-bond between DMSO and H_2O molecules. In situ optical visualization observations of the Zn/electrolyte interface during different operating conditions: (b) charging/discharging at 5 mA cm^{-2} in $6 \text{ M KOH} + 0.2 \text{ M Zn(Ac)}_2$; (c) charging/discharging at 5 mA cm^{-2} in $6 \text{ M KOH} + 2 \text{ M DMSO} + 0.2 \text{ M Zn(Ac)}_2$. (d) SEM image of the freeze-dried PAM organhydrogel electrolyte. Inset shows the corresponding optical photograph of PAM organhydrogel. (e) Ionic conductivity for the PAM organhydrogel electrolyte at various operating temperatures. (f) Long-term cycling performance of the symmetric $\text{Zn}||\text{Zn}$ battery employing PAM hydrogel electrolyte and PAM organhydrogel electrolyte at current density of 2 mA cm^{-2} .

The high-rate performance is still challenging because of the formation of dense Zn dendrites and the destruction of Zn/hydrogel electrolyte interface. Therefore, modulating the intrinsic properties of hydrogel electrolyte is particularly vital to alleviate these issues for achieving high-rate capability at ambient condition.

As known, DMSO is a favorable H-bond acceptor to form strong H-bond network with water molecules (**Figure R12a**), reconstructing the solvation sheath structure of Zn^{2+} . On the one hand, the side reaction activity in aqueous electrolytes is

obviously suppressed. In situ optical visualization observation reveals a smooth interface of Zn/electrolyte (**Figure R12b and c**) when adding DMSO into electrolyte, whereas the severe Zn dendrites appear in DMSO-free electrolyte. The PAM organhydrogel synthesized in DMSO/H₂O binary solvent systems retains a good stretchability. SEM image (**Figure R12d**) reveal an interconnected porous structure, facilitating the electrolyte trapping and fast migration of Zn²⁺ ion diffusion during electrochemical reactions. In **Figure R12e**, the ionic conductivity of the PAM organhydrogel electrolyte shows a high conductivity at room temperature (0.26 S m⁻¹) and even at -40, -60 and 60 °C (0.04, 0.0087 and 47.56 S m⁻¹, respectively), illustrating an efficient broad temperature adaptability. And the high PAM organhydrogel electrolyte retention capabilities at 20 and 60 °C are measured. In order to investigate the compatibility between Zn anode and organhydrogel electrolyte, Zn/Zn symmetric batteries were constructed. The Zn/Zn symmetric batteries with PAM organhydrogel electrolyte display a more durable stripping/plating cycles over 200 h in contrast to the PAM hydrogel electrolyte (**Figure R12f**), suggesting a stable electrochemical interface at room temperature. Meanwhile, ex XRD patterns (**Figure R13**) display the relatively weak intensity of formed Zn dendrites (ZnO) when using PAM organhydrogel electrolyte. **These results indicate that the introduction of DMSO effectively alleviates Zn dendrites and improves Zn/gel electrolyte interface stability via modulating the H-bond network of organhydrogel electrolyte.**

Figure R13. Ex situ XRD patterns of Zn plates after cycling tests under the different charging-discharging time at 20 mA cm^{-2} : (a) 1 h, (b) 2 h, (c) 4 h, (d) 6 h, (e) 8 h, and (f) 10 h.

➤ **Quasi-solid-state ZABs performance at room temperature**

Figure R14. Charging/discharging performance of quasi-solid-state ZABs with Co SA-NDGs at 50 and 100 mA cm^{-2} .

Notably, the quasi-solid-state ZABs with Co SA-NDGs using PAM organhydrogel electrolyte present a robust charging/discharging cycle at 50 and 100

mA cm⁻² (Figure R14). Moreover, the structural stability and integrity of the cathodes after the long-term cycling are confirmed (Figure R15 and R16). A statistical big data analysis (Figure R17 and Table R5) highlights this groundbreaking result at high cycling rates, implying the promising application.

Figure R15. (a) SEM image, (b) TEM image for Co SA-NDGs after the long-term cycling.

Figure R16. (a) XPS survey spectra, (b) Co 2p spectra, (c) N 1s spectra, (d) C 1s spectra and (e) O 1s spectra for Co SA-NDGs after the long-term cycling.

Figure R17. Big data analysis of cycling performance distribution for the reported quasi-solid-state ZABs measured at room temperature.

Table R5. Performance comparison of reported quasi-solid-state ZABs.

Catalysts	Gel electrolyte	OCV (V)	Peak power density (mW cm ⁻²)	Cycling time (h) @current density (mA cm ⁻²)	Reference
Co SA-NDGs	PAM	1.43	219.9	50@100 100@50	This work
Co SA@NCF/CNF	PVA	1.41		9@3	Adv. Mater. 2019 , 31 , 1808267.
NiCo _{2.148} O ₄ PNS	PVA	1.30		20@1	Adv. Mater. 2020 , 32 , 2001651.
Ce-LaCoO ₃	PVA	1.33	31	8@2	Nano Energy 2020 , 50 , 691.
CoN ₄ /NG	PVA		28	6@3	Nano Energy 2018 , 50 , 691.
CoSA/N,S-HCS	PVA	1.48		17@5	Adv. Energy Mater. 2020 , 10 , 2002896.
Fe-NC SAC	PVA	1.424	45	1@2	J. Mater. Chem. A 2020 , 8 , 9981.
Mn-CoN	PAAm	1.37	48	20@2	Sci. China Chem. 2020 , 63 , 7.
Mn ₃ O ₄ /NiCo ₂ S ₄	PVA-PEO	1.427		16.8@1	J. Power Sources 2020 , 462 , 228162.
NiCo ₂ O ₄ /MXene	PVA	1.40	55.1	33@1	ACS Appl. Mater. Interfaces 2020 , 12 , 44639.
NPF@CNF-800	PVA	1.33	64	20@5	ACS Appl. Mater. Interfaces 2021 , 13 , 13328.
CoS/CoO@NGNs	PVA	1.3	39.3	10@1	Nano-Micro Lett. 2021 , 13 , 3.
NCNTM	PAAs	1.49	176	30@1	J. Energy Chem. 2021 , 55 , 183.
FeCo/Se-CNT	PVA	1.41	37.5	20@5	Nano Lett. , 2021 , 21 , 2255.

CoFe/N-HCSs	PVA	1.4		10@2	Chem. Eng. J. 2021 , 407, 127961.
N-Mo-holely G	PANa	1.37	83	88@2	Appl. Catal.B: Environ. 2020 , 276, 119172.
AlFeCoNiCr	PANa	1.38	100	60@2	Appl. Catal. B: Environ. 2020 , 268, 118431.
FePc@N,P-DC	PVA	1.33		8@2	Appl. Catal. B: Environ. 2020 , 260, 118198.
S-Ni ₃ FeN/NSG	PAAAs	1.38	140.1	35@1	Appl. Catal. B: Environ. 2020 , 274, 119086.
Fe _{0.5} Ni _{0.5} @N-GR	PVA	1.352		18@10	Adv. Funct. Mater. 2018 , 28, 1706928.
N-NiSe ₂ /CC	PVA		27	30@1	Chem. Eng. J. 2020 , 401, 126088.
FeP/Fe ₂ O ₃ @NPCA	PVA	1.42	40.8	8@5	Adv. Mater. 2020 , 32, 2002292.
FeCo/Co ₂ P@NPCF	PVA	1.26		15@2	Adv. Energy Mater. 2020 , 10, 1903854.
Co ₃ O ₄ @x HoNPs@HPNCS	PVA	1.459	94.1	18@3	Angew. Chem. Int. Ed. 2019 , 58, 13840.
GNCNTs	PAA	1.54	223	24@1	Adv. Funct. Mater. 2019 , 30, 1906081.
Co-NC@Al ₂ O ₃	PAM	1.41	72.4	10@20	Adv. Mater. 2018 , 30, 1805268.
N-GQDs/NiCo ₂ S ₄ /CC	PVA	1.406	26.2	12@20	Small 2019 , 15, 1903610.
Meso-CoNC@GF	PVA	1.40	85.6	12@20	Adv. Mater. 2017 , 30, 1704898.
NC-Co ₃ O ₄ -90	PAM	1.44	82	21@5	Adv. Mater. 2017 , 29, 1704117.
N, S-CC	PVA-PEO	1.247	47	8@5	Adv. Sci. 2018 , 5, 1800760.
CNT@POF	PVA	1.39	22.3	4@1	Energy Environ. Sci. 2018 , 11, 1723.
Co ₃ O ₄ /CC	PVA-SiO ₂	1.27	62.6	48@3	Nano Energy 2019 , 56, 454.
CMO/S	PVA-PEO	1.32		10@1	Adv. Energy Mater. 2018 , 8, 1800612.
NGM-Co	PVA	1.439	28	1@1	Adv. Mater. 2017 , 29, 1703185.

NP-Co ₃ O ₄ /CC	PVA	1.349	99.8	20@5	Energy Storage Mater. 2020 , 26, 157.
Ni _{0.2} Co _{0.8} Se	PVA	1.428	41	8@2	Nano-Micro Lett. 2019 , 11, 28.
Co ₃ O ₄ /N-CNT	PVA	1.31		20@2	Small 2017 , 13, 1700518.
SilkNC/KB	PVA		32.3	10@1	Chem. Mater. 2019 , 31, 1023.
Co/ZnCo ₂ O ₄ @NC-CNTs	PVA	1.30	151	21@5	Nano Energy 2021 , 82, 105710.
D-CMO	PANa	1.46	149	34@2	Nano Energy 2021 , 85, 106020.
Co ₃ O ₄ /Mn ₃ O ₄ /CN _x @CNFs	PVA	1.51	191	9@1	Electrochim. Acta 2020 , 344, 136145.
HCA-Co	PAA	1.40	44.8	40@1	Chem. Eng. J. 2019 , 369, 988.
CoNC	PAAS	1.33	117	21@2	Chem. Eng. J. 2021 , 404, 127112.
Fe-N-C-700	PAA	1.42	70		Chem. Eng. J. 2021 , 405, 125956.
NiFe@N-CFs	PVA	1.18		10@1	J. Mater. Chem. A 2020 , 8, 13725.
Co-Fe-S@NSRPC	PAM-PAA	1.42	78	50@5	Nanoscale 2020 , 12, 11746.
Mn ₃ O ₄ /NiCo ₂ S ₄	PVA/PEO	1.43		16.8@1	J. Power Sources 2020 , 462, 228162.
Co/Co-N-C	PVA	1.41		10@2	Adv. Mater. 2019 , 31, 1901666.
Co-NCNT	PANa	1.45	144.6	75@2	Energy Storage Mater. 2019 , 20, 234.
Co-N _x -YSC-600	PVA	1.35	55.3	4@5	Nano Energy 2021 , 89, 106314.
WN-Ni@N,P-CNT	PVA	1.57	100.4	10@1	Appl. Catal. B: Environ. 2021 , 298, 120511.
Fe ₁ Co ₁ -CNF	PVA			3@2	Nano Energy 2021 , 87, 106147.
Co/CoO@NSC	PANa	1.43	82.7	143@2	J. Energy Chem. 2022 , 64, 385.
NiFe/N-CNT	PVA	1.41	105.4	10@1	Nano Energy 2020 , 68, 104293.

OCNT	PAA	1.39	103	17@5	Energy Storage Mater. 2020 , 30, 138.
FeCo/N-CNTs@CC	PVA	1.40	127	8@20	ACS Sustainable Chem. Eng. 2021 , 9, 4498.
FeS ₂ -CoS ₂ /NCFs	PVA	1.39	69	20@1	J. Power Sources 2021 , 482, 228955.
CoNC-MOG	PVA	1.41	63	12@2	Appl. Surf. Sci. 2021 , 537, 147818.
NBSCF	CNF-based membrane	1.44		15@1	J. Mater. Chem. A 2019 , 7, 24231.
CoFe@NCNT/CFC	PVA	1.43	37.7	15@1	J. Mater. Chem. A 2020 , 8, 18162.
Ni _{5.7} Ru _{0.3}	PVA	1.33	98.3	68@2	Chem. Commun. 2020 , 56, 13615.
NC-Co/CoN _x	PAA	1.40	41.5	25@1	Energy Storage Mater. 2019 , 16, 243.
NO-G@CP	PVA	1.33	65.1	6@5	J. Mater. Chem. A 2020 , 8, 11202.
CoO-NSC	PAA		65	35@1	ACS Appl. Mater. Interfaces 2019 , 11, 16720.
P-O/FeN ₄ -CNS	PVA	1.41	109	20@10	ACS Appl. Mater. Interfaces 2019 , 11, 33054.
CoFe/FeNC	PAA	1.47	108.6	18@2	ACS Sustainable Chem. Eng. 2020 , 8, 9009.
FeCo-N-C-700	PVA	1.43		62@10	J. Mater. Chem. A 2020 , 8, 9355.
FeN _x /N,S-C	PANa		70.6	52@5	Carbon 2020 , 166, 64.
Fe-NC SAC	PVA	1.42	45	1@1	J. Mater. Chem. A 2020 , 8, 9981.
N-CuCoS _{1.97} NWs	PVA	1.36		8@1	J. Energy Chem. 2019 , 34, 1.
CoFeP@C	PVA		72.6	20@1	ACS Appl. Mater. Interfaces 2021 , 13, 22282.
CoIn ₂ Se ₄	PVA	1.37	107	68@10	ACS Appl. Mater. Interfaces 2020 , 12, 8115.
V-Co ₃ O ₄	PVA	1.39	40.6	32@7	ACS Catal. 2021 , 11, 8097.

N-doped NiCo ₂ O ₄	PVA		23	53@5	ACS Appl. Energy Mater. 2019 , 2, 2296.
h-FeCo alloy/NCNFs	PAA	1.34	12.6	19@1	Sustainable Energy Fuels 2020 , 4, 1747
BFC-FC-0.2	PAA	1.49	160	110@2	Angew. Chem. Int. Ed. 2020 , 59, 4793.
FeNi SAs/NC	PAA	1.45	42.2	9@3	Adv. Energy Mater. 2021 , 11, 2101242.
Fe/Fe ₃ C@NdC-NCs	P-(AM-co-AA)	1.43	60	40@5	J. Mater. Chem. A 2019 , 7, 17581
FeCu-N-HC	PVA	1.41	113	110@10	Adv. Funct. Mater. 2020 , 31, 2006533.
Fe-N-C	PANa-cellulose	1.48	108.6	110@5	Adv. Energy Mater. 2019 , 9, 1803046.
CoNCNTF/CNFs	PVA	1.30	63	11@0.5	Carbon 2019 , 142, 379e387.
NiS ₂ /CoS ₂	PVA		101	33@10	J. Power Sources 2019 , 437, 226893.
Pt/RuO ₂ /CF	PAM-CNF/KOH/KI	1.45	65	75@2	Energy Storage Mater. 2021 , 42, 88.
Co ₃ O ₄ /CC	KI-PVAA-GO GPE		78.6	200@2	Adv. Mater. 2020 , 32, 1908127.
Ni-Co ₉ S ₈ /rGN	PAA	1.37	110	25@1	Appl. Catal. B: Environ. 2021 , 298, 120539.
NiCo ₂ O ₄ @N-CNWs	PVA-PEO	1.28		45@1	Electrochim. Acta 2019 , 319, 1e9.
CoSe ₂ -NCNT NSA	PAA	1.37	51.1	5@2	Nanoscale 2021 , 13, 3019.
P-CoSe ₂ /N-C FAs	PAA	1.30		20@1	Adv. Funct. Mater. 2018 , 28, 1804846.
PdNi/Ni@N-C	PAA	1.40	66.5	15@1	Energy Storage Mater. 2021 , 42, 118.

ODAC-CoO-30	PVA	1.41	42	12@2	Adv. Funct. Mater. 2021 , 31 , 2101239.
Co-NDC	PVA		45.9	3.3@2	Sci. Bull. 2018 , 63 , 548.

➤ **Low-temperature quasi-solid-state ZABs performance**

Figure R18. (a) DSC curve of the PAM organhydrogel electrolyte. Inset shows the corresponding photographs recoded at 25, -40 and -60 °C. (b) The E_b of W-W, PAM-W and A-PAM-W.

Furthermore, considering the practical demand of rechargeable batteries in cold regions, highland, etc., there is an urgent need to develop extreme-low-temperature (<-40 °C) quasi-solid-state ZABs. As far as we know, ZABs operating below -40 °C is seldomly reported owing to the pronounced increase in interfacial and charge-transfer resistance when operating temperature drops from 25 to -60 °C. The slow ion transport in extreme-low-temperature environment limits the depth of discharge and leads to low critical current density. Then, the temperature-tolerance abilities of PAM organhydrogel electrolyte are further rationalized by the differential scanning calorimetry (DSC) and dynamic mechanical analysis (DMA). As shown in **Figure R18a**, the PAM organhydrogel electrolyte remains transparent at -40 °C and further transforms into an opaque slurry gel at -60 °C. DSC curve shows that the freezing point of the PAM organhydrogel electrolyte is less than -70 °C without the emergence of an exothermic peak. Apart from the strong inter-molecular H-bonds between DMSO and H₂O, the binding energy (E_b) between the water and terminal group of PAM organhydrogel electrolytes also contributes to lower the solid-liquid transition point. In **Figure R18b**, although the E_b (-0.198 eV@PAM-W) of terminal acylamino group with neighboring water molecules via the dipole-dipole interaction is slightly

higher than that of two water molecules (-0.171 eV@W-W), the E_b of alkylated acylamino group with water molecule (A-PAM-W) increases substantially to -0.344 eV . The stronger E_b represents the lower freezing point and the better ion migration rate. The symmetric Zn/Zn battery with the PAM organhydrogel electrolyte exhibits a stable Zn plating/stripping process over 500 h at $-60 \text{ }^\circ\text{C}$ (**Figure R19**). A rough surface and dendrite-free morphology of the cycled Zn plate contribute to cycling stability. These results are indicative of anti-freezing ability for the as-synthesized PAM organhydrogel electrolyte.

Figure R19. (a) Cycling performance ($@0.5 \text{ mA cm}^{-2}$) of the symmetric Zn||Zn battery employing the PAM organhydrogel electrolyte at $-60 \text{ }^\circ\text{C}$. (b,c) SEM images of Zn plate after 500 h cycling test. Inset in (b) shows the corresponding XRD pattern. EDS result of Zn plate after 500 h cycling test. Inset shows the corresponding atomic ratio.

At $-40 \text{ }^\circ\text{C}$, the dominant factor of discharge behavior changes from ohmic polarization to concentration polarization with the increase of voltage. The quasi-solid-state ZABs using Co SA-NDGs delivers a maximum power density of 21.9 mW cm^{-2} (**Figure R20a**). Therefore, a large voltage driving force is required in ultra-low-temperature environment to reach similar current density level measured in ambient

condition. **Figure R20b** shows the rate performance of quasi-solid-state ZABs with Co SA-NDGs at -40 °C. After twice current density fluctuation tests, the discharging voltage hardly decays. The critical current density of the assembled quasi-solid-state ZABs is 2 mA cm⁻² under steady-state discharge test. The specific capacity still reaches 778.4 mAh g⁻¹ at 2 mA cm⁻² at -40 °C (**Figure R20c and d**), corresponding to an energy density of 918.5 Wh kg⁻¹. A comparison between the energy density and operating temperature of the fabricated quasi-solid-state ZABs and other low-temperature batteries previously reported (**Figure R20e and Table R6**), suggests the intrinsic advantage of ZABs for low-temperature energy storage. The exceptional charging/discharging cycling performance with high-capacity retention over 90% are recorded at different current density in **Figure R20f**.

With further decrement to -60 °C, the discharge voltages at 0.1, 0.5 and 1.0 mA cm⁻² are 1.30, 1.25 and 1.18 V (**Figure R21a**), respectively. **Figure R21b** exhibits the charging/discharging cycles at 0.5 and 1.0 mA cm⁻². The long-cycle durability with capacity retention over 90% is achieved at -60 °C. **To the best of our knowledge, this is the record of lowest operation temperature of ZABs reported so far.**

Figure R20. (a) Galvanostatic discharge curves and the corresponding power density curves of the quasi-solid-state ZABs with Co SA-NDGs at $-40\text{ }^{\circ}\text{C}$. (b) Rate performance of the Co SA-NDGs-based quasi-solid-state ZABs. (c) Galvanostatic discharge voltage platform and (d) specific capacity of the quasi-solid-state ZABs with Co SA-NDGs at 2 mA cm^{-2} . (e) Ragone plots for assembled quasi-solid-state ZABs' energy density and operating temperature with reported low-temperature solid-state batteries previously reported. (f) Charging/discharging cycling performance of quasi-solid-state ZABs with Co SA-NDGs at $-40\text{ }^{\circ}\text{C}$.

Figure R21. (a) Rate performance of the Co SA-NDGs-based quasi-solid-state ZABs measured at -60 °C. (b) Charging/discharging cycling performance of quasi-solid-state ZABs with Co SA-NDGs at different current density measured at -60 °C.

Table R6. Performance comparison of low-temperature electrochemical device.

Energy devices	Electrolyte	Working temperature (°C)	Performance	Reference
Zinc-air battery	PAM-based organohydrogel electrolyte	-60	300 h@0.5 mA cm ⁻² 100 h@1.0 mA cm ⁻²	This work
Zinc-air battery	PAM-based organohydrogel electrolyte	-40	21.9 mW cm ⁻² ; Specific capacity of 778.4 mAh g ⁻¹ ; Energy density of 918.5 Wh kg ⁻¹ ; 160 h@1.0 mA cm ⁻²	This work
Zinc-air battery	CsOH-based electrolyte	-10	57.9 mW cm ⁻² ; 160 cycles@5.0 mA cm ⁻² ; 65 cycles@10.0 mA cm ⁻² at -10 °C; 6.5 mW cm ⁻² at -40 °C	Angew. Chem. Int. Ed. 2021 , 60, 15281.
Zinc-air battery	PAA hydrogel electrolyte	-20	80.5 mW cm ⁻² ; Specific capacity of 691 mAh g ⁻¹ ; Energy density of 798 Wh kg ⁻¹	Angew. Chem. Int. Ed. 2020 , 59, 4793.
Zinc-air battery	CBCs super-ion conductors	-20	2 h of discharge at -20 °C	Nat. Energy 2021 , 6, 592.
Zinc-air battery	PAM/PAA organohydrogel electrolyte	-20	1.44 V of OCV for 0.5 h at -20 °C; 10 h@1 mA cm ⁻²	ACS Sustainable Chem. Eng. 2020 , 8, 11501.
Zinc-air battery	PAMC	-20	35.8 mW cm ⁻² ; 190 cycles@2 mA cm ⁻² at -30 °C	Chem. Eng. J. 2021 , 417, 129179.
Zinc-air battery	A-PAA hydrogel electrolyte	-30	63.6 mW cm ⁻² ; Specific capacity of 699 mAh g ⁻¹ ; Energy density of 789 Wh kg ⁻¹ ; 500 cycles@2 mA cm ⁻² at -30 °C	Energy Environ. Sci. 2021 , 14, 4926.
Zinc-air battery	PVA organohydrogel	-35	1.25 V of OCV for 120 h at -35 °C; 8.2 mW cm ⁻² ;	Adv. Mater.

	electrolyte			2020 , 32, 2001651.
Zinc-air battery	PAM-CNF/KOH/KI	-40	10 mW cm ⁻² Specific capacity of 743 mAh g ⁻¹ ; 45 h@2 mA cm ⁻²	Energy Storage Mater. 2021 , 42, 88.
Zinc-air battery	SP-DN hydrogel electrolyte	-50	1.38 V of OCV; 97 mW cm ⁻² ; Specific capacity of 620 mAh g ⁻¹	Energy Environ. Sci. 2021 , 14, 4451.
Zn/LiFePO ₄ hybrid battery	ZL-PAAm hydrogel electrolyte	-20	98% capacity retention upon cooling down to -20 °C; near 100% capacity retention with >99.5%; Coulombic efficiency over 500 cycles at -20 °C	Adv. Funct. Mater. 2019 , 30, 1907218.
Strain sensors	MXene nanocomposite organohydrogel	-40	A relatively broad strain range (up to 350% strain) and a high gauge factor of 44.85 at -40 °C	Adv. Funct. Mater. 2019 , 29, 1904507.
Super- capacitors	PVA organohydrogel	-40	70.6% capacitance retained at -40 °C and 11.7% capacitance decay over 5000 charge/discharge cycles at -20 °C	Adv. Energy Mater. 2018 , 8, 1801967.
Li metal battery	1M LiFSI DEE	-60	Discharge capacity of 236 and 13 mAh g ⁻¹ at -40 and -60 °C	Nat. Energy 2021 , 6, 303.
Li metal battery	5M LiTFSI/EA + DCM	-70	High energy (178 Wh kg ⁻¹ and power (2877 W kg ⁻¹) at -70 °C	Angew. Chem. Int. Ed. 2019 , 58, 5623.
Li metal battery	1 M LiPF ₆ MTFP/FEC (9:1)	-60	Specific capacities of 161, 149, and 133 mAh g ⁻¹ at -40, -50, and -60 °C	ACS Energy Lett. 2020 , 5, 1438.
Na metal battery	EC/PC-based electrolyte	-30	Specific capacities of 92.1 mAh g ⁻¹ at -30 °C	Energy Environ. Sci. , 2021 , 14, 4936.
Li-ion battery	EA-based electrolyte	-70	Specific capacity of 20 mAh g ⁻¹ at low rate of 0.2 C; 70% of capacity at room temperature	Joule , 2018 , 2, 902.

Na-ion battery	PHP5A electrolyte	-20	Capacity of 44.2 mAh g ⁻¹	Adv. Funct. Mater. 2019 , 30, 1906770.
Zn-ion battery	PAM polyelectrolyte	-20	Specific capacity of 160.3 mAh g ⁻¹ and stable 600 cycles at 0.2 A g ⁻¹	Energy Storage Mater. 2022 , 44, 517.
Zn-ion battery	3 M Zn(CF ₃ SO ₃) ₂	-20	Specific capacity of 120, 96, 64, and 41 mAh g ⁻¹ at 0.1, 0.2, 0.5, and 1.0 A g ⁻¹	J. Power Sources 2019 , 441, 227192.
Zn ion hybrid capacitor	Zn(ClO ₄) ₂ salty ice	-50	74.1% of the room temperature capacity at -60 °C; 280 days at 1 A g ⁻¹ at -30 °C	Adv. Funct. Mater. 2021 , 31, 2101277.
PANI LTE Zn battery	ZnCl ₂ -based electrolyte	-70	Specific capacity of 84.9 mAh g ⁻¹ and stable during over 2000 cycles with ~100% capacity retention	Nat. Commun. 2020 , 11, 4463.
Li-CO ₂ battery	DOL-based electrolyte	-60	Discharge capacity of 8976 mAh g ⁻¹ and long lifespan of 150 cycles (1500 h) with a fixed 500 mAh g ⁻¹ capacity per cycle	Adv. Funct. Mater. 2020 , 30, 2001619.
Proton battery	62 wt% (9.5 M) H ₃ PO ₄	-78	Stable cycle life for 450 cycles, high round-trip efficiency of 85%	Adv. Energy Mater. , 2020 , 10, 2000968.
Proton battery	2 M H ₂ SO ₄ + 2 M MnSO ₄	-70	Discharge capacity of 171.8 mAh g ⁻¹ and 100 cycles at 0.1 A g ⁻¹	ACS Energy Lett. 2020 , 5, 685.
Zn battery	2M Zn(CF ₃ SO ₃) ₂	-30	Specific capacity of 285.0 mAh g ⁻¹ at -30 °C and capacity retention of 81.7% after 1000 cycles	ACS Energy Lett. , 2021 , 6, 2704.
Li-S battery	AMDS-modified electrolyte	-40	Specific capacity of 2408 mAh g ⁻¹ at -30 °C and stable 50 cycles	ACS Nano 2021 , 15, 13847.
Zn-MnO ₂ battery	EG-waPUA/PAM hydrogel	-20	Specific capacity of 226 mAh g ⁻¹ at -20 °C and capacity retention of 87.41% over 600 cycles	Energy Environ. Sci. , 2019 , 12, 706.

➤ **Quasi-solid-state ZABs performance at 60 °C**

Additionally, the strong inter-molecular H-bonds between DMSO and H₂O result in a decrease in the saturated vapor pressure of H₂O molecular, preventing the evaporation of H₂O at elevated temperatures. The DMA result shows that the glass transition temperature (T_g) of the PAM organhydrogel is 125 °C (**Figure R22**). These results are indicative of thermally stable properties for the as-synthesized PAM organhydrogel electrolyte. Moreover, this quasi-solid-state ZABs also operate well at temperatures ranging from 20 to 60 °C (**Figure R23a**). At elevated temperature of 60 °C, the maximum power density is 285.7 mW cm⁻² with the average discharge voltages of 1.26 V @ 10 mA cm⁻² and 1.23 V @ 20 mA cm⁻² (**Figure R23b and c**), respectively. The remarkable cycling stability is also recorded without significant decay after 60 h (**Figure R23d**).

Figure R22. DMA analysis of PAM organhydrogel under temperatures ranges from 25 to 200 °C.

Figure R23. (a) Charging/discharging curves and (b) power density curves of quasi-solid-state ZABs with Co SA-NDGs at different temperatures. (c) Galvanostatic discharge voltage platforms at 10 and 20 mA cm^{-2} . (d) Charging/discharging cycling performance of quasi-solid-state ZABs with Co SA-NDGs measured at 60 °C.

Conclusively, based on experimental evidences and theoretical calculations, we conclude the beneficial effect of graphene curvature on atomic Co-N₄-C system for optimizing the ORR activity. The quasi-solid-state ZABs with Co SA-NDGs not only achieve the record-high cycling rates of 100 mA cm^{-2} over 50 h at room temperature, but also show exceptional cycling stability with capacity retention over 90% more than 300 h (0.5 mA cm^{-2}) at -60 °C. For the previously reported ZABs, our results are undoubtedly a major performance breakthrough toward practical applications as needed. Besides, we also have detailed analysis

of PAM organhydrogel electrolyte. The introduction of DMSO into PAM hydrogel electrolyte effectively modulates the H-bond network, enabling improved interface stability and all-temperature adaptability (from -60 to 60°C) simultaneously. Based on the above discussion, we believe our work would be of broad interest to *Nature Communications*'s readers and promote the research in electrocatalysts with curved nanostructure and quasi-solid-state ZABs.

2. In rate performance and discharge-charge cycling, the test time (10 min) for each step is too short which is not enough for practical applications. The cycling performance of long discharge/charge time (1-4 hours) for each cycle should be added.

Response: Thank you very much for your professional advice. Following your advice, we added the cycling performance of long discharge/charge time at 5 mA cm⁻² (every cycle of 1h). As seen, the Co SA-NDGs-based aqueous ZABs also display a robust discharging/charging stability over 360 h with significant voltage loss (**Figure R24b**).

Figure R24. Charging/discharging cycling performance of aqueous ZABs at current density of (a) 10 mA cm⁻² (every cycle of 22 min) and (b) 5 mA cm⁻² (every cycle of 1 h).

The relevant content has been added to the revised Manuscript and Supplementary

Information (Supplementary Figure 22b), which were marked in yellow.

Figure R25. Galvanostatic discharge curves of the aqueous ZABs with Co SA-NDGs at 100 mA cm⁻².

Meanwhile, the discharge curves of the aqueous ZABs with Co SA-NDGs at 100 mA cm⁻² over 240 h is displayed in **Figure R25**. Such a long time of discharge confirms its application potential.

3. One key highlight of this manuscript is the adaptability of hydrogel electrolytes in a large range of working temperatures. Therefore, the retention of H₂O in organ-hydrogel electrolytes, under different temperatures, as an important characterization, should be analyzed.

Response: Thank you very much for your professional advice. Large amounts of water in hydrogel electrolytes inevitably freeze and restrict ion transport at subzero temperatures, and the water molecules are unable to remain under elevated temperatures. Because the zinc-air battery is a distinctive “half-open” device that utilizes the air, therefore, the water retention capability of PAM organhydrogel

electrolytes is an important parameter. **For low-temperature battery working at subzero temperatures, the retention of H₂O is close to 100%, mainly due to negligible water evaporation. At 20 °C, the PAM organhydrogel electrolytes could only maintain 87.2% of the initial weight after a week without any sealing treatment (Figure R26). At 60 °C, a high 72.0% of water retention capability is obtained in a 56 h test.** Compared with other gel electrolyte (Table R7),^[1-10] it is concluded that the PAM organhydrogel electrolyte has an efficient water retention capability. The strong inter-molecular H-bonds between DMSO and H₂O result in a decrease in the saturated vapor pressure of H₂O molecular, preventing the evaporation of H₂O at elevated temperatures.

Figure R26. Electrolyte retention capabilities of the of PAM organhydrogel electrolyte at (a) 20 °C and (b) 60 °C.

Electrolyte-retention capability measurement: The pristine PAM organhydrogel electrolyte was cutted into cylinders with diameter of 30 mm and weighed. Then, the sample was completely exposed to the air at 20 and 60 °C. The weight was recorded at different times. The percentage of residual weight (W_r) was calculated by the following formula:

$$W_r (\%) = W_t/W_0*100\%$$

where W_t and W_0 represent the weight at a certain time and the initial weight, respectively.

Table R7. Electrolyte retention capability comparison of used hydrogel electrolyte.

Hydrogel electrolyte	Electrolyte retention capability (%)	Reference
PAM organhydrogel electrolyte	87.2%@168 h (20 °C) 72%@56 h (60 °C)	This work
PVA	64.7%@12 h (25 °C)	Adv. Mater. 2020 , 32, 1908127.
KI-PVAA-GO	90%@12 h (25 °C)	Adv. Mater. 2020 , 32, 1908127.
G-CyBA/PAAm SP-DN	99%@12 h (25 °C) 88%@12 h (60 °C)	Energy Environ. Sci. 2021 , 14, 4451.
PAM/PAA hydrogels	19.1%@48 h (25 °C)	ACS Sustainable Chem. Eng. 2020 , 8, 11501.
P-(AM-co-AA) alkaline gels	84%@100 h (25 °C)	J. Mater. Chem. A 2019 , 7, 17581.
PANa hydrogel	63.2%@240 h (25 °C)	Energy Storage Mater. 2019 , 20, 234.
PVA-5 wt.% SiO ₂ nanocomposite GPEs	62.4%@40 h (25 °C)	Nano Energy 2019 , 56, 454.
PAMPS/PAAm OHE	62.4%@40 h (25 °C)	Small Methods 2022 , 6, 2101043.
PANa hydrogel electrolyte	97.3%@185 h (25 °C)	Adv. Energy Mater. 2018 , 8, 1802288.
PAM-CNF/KOH/KI	86%@12 h (25 °C)	Energy Storage Mater. 2021 , 42, 88.
K-PAA hydrogel electrolyte	60%@240 h (25 °C)	Angew. Chem. Int. Ed. 2020 , 59, 4793.

4. In Figure 4, the charging/discharging cycling performance at a low current density at -60 °C was displayed. How about cycling performance at high current density (50 mA cm⁻² or 100 mA cm⁻²) at -60 °C?

Response: Thank you very much for your professional advice. The ionic diffusion rate of organhydrogel electrolyte is significantly lowered in low-temperature environment, leading to a considerable resistance enhancement (**Figure R27**) and large battery polarization. The achieved critical current density in low-temperature is severely limited compared to that of ambient condition. In our work, we used current density of 0.5 and 1.0 mA cm⁻² to assess cycling performance at -60 °C. At 2.0 mA cm⁻², the charging/discharging overpotential shows a significant enhancement, displaying a severely battery polarization (**Figure R28**). As high current density, the quasi-solid-state ZABs cannot operate, which is also a common problem of all low-temperature batteries. Therefore, **improving the ionic diffusion rate and decreasing the interfacial transport resistance is necessary to enable high current density in low-temperature environment, which is also the direction of our further efforts.**

Figure R27. EIS results of the quasi-solid-state ZABs with Co SA-NDGs measured at (a) 25°C, (b) -40°C and (c) -60°C. (d) Equivalent circuit diagram.

Figure R28. Charging/discharging cycling performance of quasi-solid-state ZABs with Co SA-NDGs at 2.0 mA cm^{-2} measured at $-60 \text{ }^{\circ}\text{C}$.

Reference

1. Z. Song, J. Ding, B. Liu, X. Liu, X. Han, Y. Deng, W. Hu, C. Zhong, A rechargeable Zn-air battery with high energy efficiency and long life enabled by a highly water-retentive gel electrolyte with reaction modifier. *Adv. Mater.* **32**, 1908127 (2020).
2. C. Gu, X. Xie, Y. Liang, J. Li, H. Wang, K. Wang, J. Liu, M. Wang, Y. Zhang, M. Li, H. Kong, C. S. Liu, Small molecule-based supramolecular-polymer double-network hydrogel electrolytes for ultra-stretchable and waterproof Zn-air batteries working from -50 to $100 \text{ }^{\circ}\text{C}$. *Energy Environ. Sci.* **14**, 4451-4462 (2021).
3. R. Chen, X. Xu, S. Peng, J. Chen, D. Yu, C. Xiao, Y. Li, Y. Chen, X. Hu, M. Liu, H. Yang, I. Wyman, X. Wu. A flexible and safe aqueous zinc-air battery with a wide operating temperature range from -20 to $70 \text{ }^{\circ}\text{C}$. *ACS Sustainable Chem. Eng.* **8**, 11501-11511 (2020).
4. Z. Cao, H. Hu, M. Wu, K. Tang, T. Jiang, Planar all-solid-state rechargeable Zn-air batteries for compact wearable energy storage. *J. Mater. Chem. A* **7**, 17581-17593

- (2019).
5. Z. Pei, Y. Huang, Z. Tang, L. Ma, Z. Liu, Q. Xue, Z. Wang, H. Li, Y. Chen, C. Zhi, Enabling highly efficient, flexible and rechargeable quasi-solid-state zn-air batteries via catalyst engineering and electrolyte functionalization. *Energy Storage Mater.* **20**, 234-242(2019).
 6. X. Fan, J. Liu, Z. Song, X. Han, Y. Deng, C. Zhong, W. Hu, Porous nanocomposite gel polymer electrolyte with high ionic conductivity and superior electrolyte retention capability for long-cycle-life flexible zinc-air batteries. *Nano Energy* **56**, 454-462 (2019).
 7. D. Jiang, H. Wang, S. Wu, X. Sun, J. Li, Flexible zinc-air battery with high energy efficiency and freezing tolerance enabled by DMSO-based organohydrogel electrolyte. *Small Methods* **6**, 2101043 (2022).
 8. Y. Huang, Z. Li, Z. Pei, Z. Liu, H. Li, M. Zhu, J. Fan, Q. Dai, M. Zhang, L. Dai, C. Zhi, Solid-state rechargeable Zn//NiCo and Zn-air batteries with ultralong lifetime and high capacity: the role of a sodium polyacrylate hydrogel electrolyte. *Adv. Energy Mater.* **8**, 1802288 (2018).
 9. Y. Zhang, H. Qin, M. Alfred, H. Ke, Y. Cai, Q. Wang, F. Huang, B. Liu, P. Lv, Q. Wei, Reaction modifier system enable double-network hydrogel electrolyte for flexible zinc-air batteries with tolerance to extreme cold conditions. *Energy Storage Mater.*, **42**, 88-96 (2021).
 10. Z. Pei, Z. Yuan, C. Wang, S. Zhao, J. Fei, L. Wei, J. Chen, C. Wang, R. Qi, Z. Liu, Y. Chen, A flexible rechargeable zinc-air battery with excellent low-temperature adaptability. *Angew. Chem. Int. Ed.* **59**, 4793-4799 (2020).

Finally, we sincerely thank you for your valuable and sincere suggestions. According to your suggestions, we believe that the revised manuscript will be more scientific and rigorous.

Reviewer #3 (Remarks to the Author):

The authors reported an interesting quasi-solid-state Zn-air battery with a high cycling rate and high-capacity retention. The capacity retention is remarkable even under -60 °C, due to the formation of hydrogen bonds between the solvent DMSO and PAM hydrogel. Both experimental and theoretical work had been carried out to characterize the performance of the battery. The results are interesting, but the manuscript should be revised and checked carefully since there are quite a few grammatical mistakes. Here I list a few of them.

1. In the abstract, the sentence “the modulated H-bond network of polyacrylamide (PAM) organohydrogel electrolyte with” should be checked.

Response: Thank you very much for your professional advice. This is our mistake. We have now corrected this mistake. Therefore, the complete sentence in the revised Manuscript is as follows: **Here we present an ultrahigh-rate and robust quasi-solid-state ZABs integrated with the atomic Co-N₄ sites anchored on wrinkled nitrogen-doped graphene (Co SA-NDGs) cathode and the polyacrylamide (PAM) organohydrogel electrolyte with the modulated H-bond network.**

2. On page 11: “Density of states (DOS) analysis (Fig. 2h) show that” and “charge transfer difference”.

Thank you very much for your professional advice. This is our mistake. We have now corrected this mistake. Therefore, the revised sentence in the revised Manuscript is as follows: **The curved Co-N₄ shows more localized charge densities and a larger charge gradient than the planar Co-N₄, which is believed to facilitate the subsequent O₂ activation. Density of states (DOS) analysis (Fig. 2h) shows that the d-band center of curved Co-N₄ is downshifted compared to that of planar Co-N₄, indicating that the adsorption of oxygenated intermediates is lowered.**

3. On page 14: “reconstructing the solvation Zn^{2+} solvation sheath structure”.

Response: Thank you very much for your professional advice. This is our mistake. We have now corrected this mistake. Therefore, the revised sentence is as follows: **reconstructing the solvation sheath structure of Zn^{2+} .**

4. On page 15: “ZABs worked below $-40\text{ }^{\circ}\text{C}$ is seldomly reported”.

Response: Thank you very much for your professional advice. This is our mistake. We have now corrected this mistake. Therefore, the corrected sentence in the revised Manuscript is as follows: **ZABs operating below $-40\text{ }^{\circ}\text{C}$ is seldomly reported.**

5. On page 22: the details of the dynamics in “Molecular dynamics simulations” is missing. There were no dynamical results presented in the manuscript. If only optimization had been performed in the computational part, this section should be merged into the following section “Quantum chemistry calculations”.

Response: Thank you very much for your professional advice. This is our mistake. We have now corrected this mistake.

Quantum chemistry calculations. Our first principles calculations were based on density function theory (DFT) using the Perdew-Burke-Ernzerhof (PBE) form for the generalized gradient approximation as implemented in the Vienna ab initio simulation package code. The projector augmented wave method with a plane-wave basis set was adopted and the energy cutoff was set to 500 eV. A Monkhorst-centered $7\times 7\times 1$ k-mesh was used for structure optimization and total energy calculations. We set the vacuum space to 15 Å along the z-direction, as well as the convergence criteria for total energy and force were set to 10^{-6} eV and 0.01 eV/Å, respectively.

6. The full form should be used when you mentioned the abbreviation for the first time, such as “Vienna ab initio simulation package (VASP)” on page 23.

Response: Thank you very much for your professional advice. This is our mistake. We have now corrected this mistake. Therefore, the corrected sentence in the revised

Manuscript is as follows:

The computations were performed using the Vienna ab initio simulation package (VASP) code with PBE method.

Finally, we sincerely thank you for your valuable and sincere suggestions. According to your suggestions, we believe that the revised manuscript will be more scientific and rigorous.

Reviewer comments, second round –

Reviewer #1 (Remarks to the Author):

The authors addressed all my comments in detail. The manuscript is suitable for publication in Nature Communications.

Reviewer #2 (Remarks to the Author):

The authors have sufficiently addressed the comments of the reviewers. I have no further comments.